# CALIBRATING LLMS WITH INFORMATION-THEORETIC EVIDENTIAL DEEP LEARNING

**Yawei Li    David Rügamer    Bernd Bischl    Mina Rezaei**
Department of Statistics, LMU Munich
Munich Center for Machine Learning (MCML)

## ABSTRACT

Fine-tuned large language models (LLMs) often exhibit overconfidence, particularly when trained on small datasets, resulting in poor calibration and inaccurate uncertainty estimates. Evidential Deep Learning (EDL), an uncertainty-aware approach, enables uncertainty estimation in a single forward pass, making it a promising method for calibrating fine-tuned LLMs. However, despite its computational efficiency, EDL is prone to overfitting, as its training objective can result in overly concentrated probability distributions. To mitigate this, we propose regularizing EDL by incorporating an information bottleneck (IB). Our approach **IB-EDL** suppresses spurious information in the evidence generated by the model and encourages truly predictive information to influence both the predictions and uncertainty estimates. Extensive experiments across various fine-tuned LLMs and tasks demonstrate that IB-EDL outperforms both existing EDL and non-EDL approaches. By improving the trustworthiness of LLMs, IB-EDL facilitates their broader adoption in domains requiring high levels of confidence calibration. Code is available at `https://github.com/sandylaker/ib-edl`.

## 1 INTRODUCTION

Large language models (LLMs) have revolutionized natural language processing, with fine-tuning emerging as a prevalent method to adapt these models for specific tasks or domains (Houlsby et al., 2019; Hu et al., 2022). However, *fine-tuned* LLMs often display overconfidence in their predictions (Jiang et al., 2021; Yang et al., 2024), which compromises their reliability and limits their applicability in critical domains where trustworthiness is essential.

Overconfidence in LLMs often manifests as poor calibration, where the predicted probabilities do not accurately reflect the model's uncertainty about its predictions. Uncertainty-aware methods improve calibration by explicitly quantifying the uncertainty in the model's predictions, allowing the model to produce confidence scores that better correspond to the actual likelihood of correctness. Traditional uncertainty-aware methods, such as MC-Dropout (Gal & Ghahramani, 2016) and Deep Ensemble (Lakshminarayanan et al., 2017; Fort et al., 2019) are commonly used to mitigate overconfidence in neural networks. However, these approaches typically require multiple forward passes, significantly increasing the inference time for LLMs.

Evidential Deep Learning (EDL) (Sensoy et al., 2018; Malinin & Gales, 2018) offers a more efficient alternative by providing uncertainty estimates with a single forward pass. Despite its success in various tasks, recent studies (Deng et al., 2023; Chen et al., 2024) indicate that EDL can still yield overconfident predictions which leads to inaccurate uncertainty estimates, degrading the model's calibration performance. This issue arises from the propensity of vanilla EDL to encourage models to generate excessive evidence (i.e., support for a class) with extremely large magnitudes, leading to overly confident predicted class probabilities.

Motivated by these challenges, we propose a novel regularization approach for EDL using an *information bottleneck*, which we term **IB-EDL**. IB-EDL adaptively distorts the evidence generated by the LLM while maximally preserving the model's performance. In doing so, IB-EDL encourages the model to suppress spurious or uninformative evidence that could lead to overconfident predictions. Our theoretical analysis shows that the information bottleneck effectively penalizes the generation

of disproportionately large evidence, thereby reducing overconfidence. Notably, our method introduces less than $2\%$ computational overhead compared to a pretrained LLM, maintaining the model's inference efficiency while significantly improving its calibration. Our contributions are as follows:

- We introduce IB-EDL, an information-theoretic framework for regularizing EDL. First, we identify a theoretical issue previously overlooked in the IB literature. In the context of EDL, we address this challenge through a novel choice of the IB stochastic variable. Moreover, our solution naturally imposes an $\ell_2$ regularization, effectively mitigating the issue of overly large evidence highlighted in prior EDL research.
- We show that several existing EDL methods can be seen as a special case within the IB-EDL framework. This unification offers a cohesive perspective on these approaches.
- We perform extensive experiments on calibrating *fine-tuned* LLMs using EDL, thereby extending its applicability beyond the medium-sized networks commonly used in the EDL literature. Our results across various LLMs and datasets demonstrate that IB-EDL scales effectively.

## 2 BACKGROUND

### 2.1 MLE FINE-TUNING OF LLMS

Let $\boldsymbol{x} \in \mathcal{V}^S$ represent the input sequence for an LLM, where $\mathcal{V}$ represents the set of tokens (vocabulary) and $S$ is the sequence length. The target space is denoted by $\mathcal{Y}$, which may be identical to $\mathcal{V}$ (e.g., in next-token prediction) or a different set (e.g., in sentiment analysis). We generally assume that $|\mathcal{Y}| = C$. As we focus on the context of LLMs, *we will use the term "tokens" instead of "classes"* throughout this paper. In tasks like next-token prediction, the target can also be a sequence of tokens. For clarity in our theoretical analysis, we focus on a single-token target, with the understanding that a token sequence can be treated as multiple single-token targets.

Let $f$ be the LLM. The output logits of the LLM, $f(\boldsymbol{x}) \in \mathbb{R}^C$, are passed through a Softmax function, yielding a vector $\boldsymbol{\pi}$ with entries $\pi_j = \exp(f(\boldsymbol{x})_j)/\sum_{j'=1}^C \exp(f(\boldsymbol{x})_{j'})$, which represent the probability for each token. Let $\boldsymbol{y} \in \{0,1\}^C$ be the one-hot encoded target. Then, $\boldsymbol{y}|\boldsymbol{x}$ conforms to a categorical distribution $p(\boldsymbol{y}|\boldsymbol{x}) = \text{Cat}(\boldsymbol{y}; \boldsymbol{\pi}) = \prod_{j=1}^C \pi_j^{y_j}$. Fine-tuning LLMs on downstream tasks typically involves minimizing $-\log p(\boldsymbol{y}|\boldsymbol{x})$ which corresponds to maximum likelihood estimation (MLE). However, fine-tuning LLMs on small downstream datasets can result in overfitting and overconfident predictions. Additionally, MLE yields a *deterministic* model that cannot express uncertainty in the predicted $\boldsymbol{\pi}$.

### 2.2 UNCERTAINTY-AWARE MODELING VIA EDL

While conventional uncertainty-aware methods like a Deep Ensemble can alleviate overconfidence and improve calibration, they require multiple forward passes during inference. This can be particularly challenging for LLMs due to their already substantial computational demands. EDL provides a more efficient alternative by capturing uncertainty in *a single forward pass*, making it especially suitable for large models. EDL builds on the principles of Subjective Logic (Jøsang, 1997; 2016), which is derived from Dempster-Shafer Theory (DST) (Dempster, 1968; Shafer, 1976).

**EDL inference pipeline:** Instead of directly predicting the token probabilities $\boldsymbol{\pi}$, EDL uses the model to predict a *Dirichlet prior* on $\boldsymbol{\pi}$. Specifically, the model's output is interpreted as *pre-evidence* $\tilde{\boldsymbol{e}} = f(\boldsymbol{x}) \in \mathbb{R}^C$, which is converted into a non-negative *evidence* vector $\boldsymbol{e} = \text{SoftPlus}(\tilde{\boldsymbol{e}})$ using the SoftPlus activation. Each element $e_j$ of the evidence vector represents the amount of support for token $j$ being the correct prediction. Once the evidence is obtained, we can proceed to predict the Dirichlet prior $\text{Dir}(\boldsymbol{\pi}; \boldsymbol{\alpha})$ over the simplex of possible token probabilities $\boldsymbol{\pi} = [\pi_1, \ldots, \pi_C]^\top$ by computing the Dirichlet parameters $\alpha_j = e_j + 1, \forall j \in [C]$. More formally,

$$p(\boldsymbol{\pi}|\boldsymbol{\alpha}) = \frac{\Gamma(\alpha_0)}{\prod_{j=1}^C \Gamma(\alpha_j)} \prod_{j=1}^C \pi_j^{\alpha_j - 1}, \quad \text{with } \alpha_0 = \sum_{j=1}^C \alpha_j, \qquad (1)$$

where $\Gamma(\cdot)$ is the *gamma* function. The expected probabilities $\hat{\pi}_j$ and the final predicted token $\hat{y}$ are:

$$\hat{\pi}_j = \mathbb{E}_{\boldsymbol{\pi} \sim \text{Dir}(\boldsymbol{\pi}; \boldsymbol{\alpha})}[\pi_j|\boldsymbol{\alpha}] = \frac{\alpha_j}{\alpha_0} = \frac{e_j + 1}{\sum_{j=1}^C (e_j + 1)}, \quad \hat{y} = \arg\max_j \hat{\pi}_j. \qquad (2)$$

In summary, the EDL pipeline can be symbolized as: $\boldsymbol{x} \to f(\boldsymbol{x}) \to \tilde{\boldsymbol{e}} \to \boldsymbol{e} \to \boldsymbol{\alpha} \to \boldsymbol{\pi} \to \boldsymbol{y}$.

**Uncertainty estimate:** EDL also enables quantifying uncertainty in the model's prediction. This is done through the concepts of *belief mass* $b_j$ and *uncertainty mass* $u$ in the Subjective Logic:

$$b_j = (\alpha_j - 1)/\alpha_0, \quad u = C/\alpha_0. \tag{3}$$

Similar to the evidence, the belief mass $b_j$ indicates the support for token $j$ being the correct prediction, while the uncertainty mass $u$ captures the model's overall uncertainty about the prediction. The sum of all belief masses and the uncertainty mass is normalized: $\sum_{j=1}^{C} b_j + u = 1$.

## 2.3 TRAINING OF EDL NETWORKS

In EDL, models are usually trained by minimizing the Bayes risk, which involves the expected loss under the Dirichlet distribution. Given the predicted $\boldsymbol{\alpha}$ from an input $\boldsymbol{x}$, and the target $\boldsymbol{y} \in \{0, 1\}^C$, the Bayes risk for the cross-entropy loss is defined as $\mathcal{L}_{\mathrm{CE}}(\boldsymbol{\theta}) = \mathbb{E}_{\boldsymbol{\pi} \sim \mathrm{Dir}(\boldsymbol{\pi};\boldsymbol{\alpha})} \left[ -\sum_{j=1}^{C} y_j \log(\pi_j) \right] = \sum_{j=1}^{C} y_j (\psi(\alpha_0) - \psi(\alpha_j))$, where $\boldsymbol{\theta}$ denotes the trainable parameters of the model, and $\psi$ is the *digamma* function. To stabilize the training, Sensoy et al. (2018) introduce the MSE loss as an alternative objective, which can be analytically computed using $\boldsymbol{\alpha}$:

$$\mathcal{L}_{\mathrm{MSE}}(\boldsymbol{\theta}) = \mathbb{E}_{\boldsymbol{\pi} \sim \mathrm{Dir}(\boldsymbol{\pi};\boldsymbol{\alpha})} \|\boldsymbol{y} - \boldsymbol{\pi}\|_2^2 = \sum_{j=1}^{C} \left( y_j - \frac{\alpha_j}{\alpha_0} \right)^2 + \frac{\alpha_j(\alpha_0 - \alpha_j)}{\alpha_0^2(\alpha_j + 1)}. \tag{4}$$

For detailed derivations, we refer to Sensoy et al. (2018). Furthermore, they introduce a regularization term to *suppress evidence for non-target tokens*, i.e., the tokens labeled as $\boldsymbol{0}$ in $\boldsymbol{y}$. This is achieved by first "removing" the evidence associated with the target token, using $\tilde{\boldsymbol{\alpha}} = \boldsymbol{y} + (\boldsymbol{1} - \boldsymbol{y}) \odot \boldsymbol{\alpha}$, where $\boldsymbol{1} = [1, \dots, 1]^\top$. The regularization term is then defined as

$$\mathcal{L}_{\mathrm{Reg}}(\boldsymbol{\theta}) = D_{\mathrm{KL}} \left( \mathrm{Dir}\left(\boldsymbol{\pi}; \tilde{\boldsymbol{\alpha}}\right) \, \| \, \mathrm{Dir}\left(\boldsymbol{\pi}; \boldsymbol{1}\right) \right), \tag{5}$$

where $D_{\mathrm{KL}}$ denotes the Kullback-Leibler (KL) divergence. The total loss for training is given by:

$$\mathcal{L}_{\mathrm{EDL}}(\boldsymbol{\theta}) = \mathcal{L}_{\mathrm{MSE}}(\boldsymbol{\theta}) + \lambda \cdot \mathcal{L}_{\mathrm{Reg}}(\boldsymbol{\theta}), \tag{6}$$

where $\lambda > 0$ is a hyper-parameter. Note that $\mathcal{L}_{\mathrm{MSE}}(\boldsymbol{\theta})$ can be replaced with $\mathcal{L}_{\mathrm{CE}}(\boldsymbol{\theta})$.

However, as the model is trained to minimize the empirical risk, it remains susceptible to overfitting the data and producing overconfident predictions. The objectives in $\mathcal{L}_{\mathrm{CE}}(\boldsymbol{\theta})$ and $\mathcal{L}_{\mathrm{MSE}}(\boldsymbol{\theta})$ drive the learned Dirichlet distribution towards a Dirac delta distribution. Consequently, the trained model may produce $\alpha_j$ with extreme magnitudes for the target token $j$ (Chen et al., 2024). Eq. (5) also does not fully address this issue, as it only suppresses the evidence of non-target tokens.

Recent efforts have sought to mitigate the overconfidence issue in EDL. For instance, I-EDL (Deng et al., 2023) incorporates the Fisher Information matrix into the distribution of $\boldsymbol{y}$. R-EDL (Chen et al., 2024) alleviates overconfidence by relaxing $\alpha_j = e_j + 1, \forall j$ to $\alpha_j = e_j + \eta, \forall j$ with a hyper-parameter $\eta \in \mathbb{R}_+$. Orthogonal to these approaches, we do not alter the assumptions on $\boldsymbol{y}$ or $\boldsymbol{\alpha}$ in the EDL formulation. Instead, we impose regularization on the model using an information bottleneck (IB), which *discourages* the model from relying on irrelevant or spurious correlations that could lead to an overly concentrated Dirichlet distribution, thereby preventing overconfident predictions.

## 3 INFORMATION BOTTLENECK-REGULARIZED EDL

We begin by adopting an information-theoretic perspective on neural networks and introduce the IB objective. We then explain how to regularize EDL with IB and why the final IB-EDL objective effectively mitigates overconfidence.

### 3.1 A CAREFUL EXAMINATION OF THE INFORMATION BOTTLENECK CRITERION

**A high-level view of IB:** Let $X$ be the input random variable and $Y$ represent the random variable of the target token. We also introduce an intermediate representation $Z$, which serves as a stochastic encoding of $X$. In the context of EDL, $Z$ can take various forms, such as the internal features

of any LLM layer, the pre-evidence $\tilde{e}$ (i.e., model output), evidence $e$, Dirichlet parameters $\boldsymbol{\alpha}$, or token probabilities $\boldsymbol{\pi}$. Choosing different forms of $Z$ corresponds to selecting features that capture different levels of abstraction. We will discuss our selection of $Z$ in Section 3.2. Since input data often contains redundant or irrelevant information, which may hinder the generalization ability of $Z$, it is essential for $Z$ to retain the most predictive information about $Y$ while discarding irrelevant information from $X$. This trade-off leads to better generalization. The Information Bottleneck method (Tishby et al., 1999; Tishby & Zaslavsky, 2015) formalizes this principle through the concept of mutual information. Specifically, the IB objective is:

$$\max_{\boldsymbol{\theta}} I(Z, Y; \boldsymbol{\theta}) - \beta I(Z, X; \boldsymbol{\theta}), \tag{7}$$

where $I(\cdot, \cdot)$ represents mutual information (MI), and $\beta > 0$ is a hyperparameter controlling the trade-off between relevance and compression. Since $Z$ is computed by the model $f$, optimizing the model's parameters $\boldsymbol{\theta}$ is equivalent to optimizing $Z$. The term $I(Z, Y; \boldsymbol{\theta})$ promotes $Z$ to be predictive of $Y$, while $I(Z, X; \boldsymbol{\theta})$ encourages $Z$ to ignore irrelevant information from $X$. For simplicity, we omit the model parameters $\boldsymbol{\theta}$ in the following equations.

The mutual information terms in Eq. (7) are generally intractable. To address this, Alemi et al. (2017) proposed to derive more tractable variational bounds. Following Wieczorek & Roth (2020), we assume the Markov chain $X - Z - Y$[1] to derive the IB objective. Detailed derivations for the following equations are provided in Appendix A.

**Upper bound of $I(Z, X)$:** To derive a variational upper bound on $I(Z, X)$, we first expand it as:

$$I(Z, X) = \int p(\boldsymbol{x}, \boldsymbol{z}) \log \frac{p(\boldsymbol{z}|\boldsymbol{x})}{p(\boldsymbol{z})} \mathrm{d}\boldsymbol{z}\mathrm{d}\boldsymbol{x} = \mathbb{E}_{p(\boldsymbol{z}|\boldsymbol{x})}\mathbb{E}_{p(\boldsymbol{x})}[\log p(\boldsymbol{z}|\boldsymbol{x})] - \mathbb{E}_{p(\boldsymbol{z})}[\log p(\boldsymbol{z})].$$

Computing $p(\boldsymbol{z}) = \int p(\boldsymbol{z}, \boldsymbol{x}) \mathrm{d}\boldsymbol{x}$ is challenging as it involves marginalizing over $\boldsymbol{x}$. Instead, Alemi et al. (2017) suggest approximating it using a predefined prior $r(\boldsymbol{z})$. We will discuss how to choose $r(\boldsymbol{z})$ in Section 3.2. By utilizing the Kullback–Leibler divergence $D_{\mathrm{KL}}\left(p(\boldsymbol{z})\|r(\boldsymbol{z})\right) \geq 0$, we obtain:

$$I(Z, X) \leq \int p(\boldsymbol{z}|\boldsymbol{x})p(\boldsymbol{x}) \log \frac{p(\boldsymbol{z}|\boldsymbol{x})}{r(\boldsymbol{z})} \mathrm{d}\boldsymbol{z}\mathrm{d}\boldsymbol{x} = \mathbb{E}_{p(\boldsymbol{x})}\left[D_{\mathrm{KL}}\left(p(\boldsymbol{z}|\boldsymbol{x})\|r(\boldsymbol{z})\right)\right]. \tag{8}$$

**Lower bound of $I(Z, Y)$:** Wieczorek & Roth (2020) derive the following lower bound:

$$\begin{aligned} I(Z, Y) &= \mathbb{E}_{p(\boldsymbol{x}, \boldsymbol{y})}\mathbb{E}_{p(\boldsymbol{z}|\boldsymbol{x}, \boldsymbol{y})}\left[\log p(\boldsymbol{y}|\boldsymbol{z})\right] + H(Y) \\ &\geq \mathbb{E}_{p(\boldsymbol{x})}\mathbb{E}_{p(\boldsymbol{y}|\boldsymbol{x})}\mathbb{E}_{p(\boldsymbol{z}|\boldsymbol{x})}\left[\log p(\boldsymbol{y}|\boldsymbol{z})\right], \end{aligned} \tag{9}$$

where $H(\cdot)$ denotes the Shannon entropy. By plugging the upper bound from Eq. (8) and lower bound from Eq. (9) into Eq. (7), and flipping the $\max$ to a $\min$, the IB objective becomes

$$\min_{\boldsymbol{\theta}} -\mathbb{E}_{p(\boldsymbol{x})}\mathbb{E}_{p(\boldsymbol{y}|\boldsymbol{x})}\mathbb{E}_{p(\boldsymbol{z}|\boldsymbol{x})}\left[\log p(\boldsymbol{y}|\boldsymbol{z})\right] + \beta \, \mathbb{E}_{p(\boldsymbol{x})}\left[D_{\mathrm{KL}}\left(p(\boldsymbol{z}|\boldsymbol{x})\|r(\boldsymbol{z})\right)\right], \tag{10}$$

where the expectations can be approximated using Monte Carlo samples. In the following paragraph and Section 3.2, we will discuss how to compute $p(\boldsymbol{y}|\boldsymbol{z})$, $p(\boldsymbol{z}|\boldsymbol{x})$, and how to select the prior $r(\boldsymbol{z})$.

**Challenges when applying IB to an internal layer of an LLM:** Previous works, such as Alemi et al. (2017), typically apply IB to an intermediate layer within the neural network, treating the features at that layer as the hidden variable $Z$. In this scenario, the earlier layers (a.k.a. the *encoder*) learn $p(\boldsymbol{z}|\boldsymbol{x})$. However, since $\boldsymbol{z}$ represents the features at an intermediate layer, the true distribution $p(\boldsymbol{y}|\boldsymbol{z})$ is unknown in practice. As a result, Alemi et al. (2017) use the later layers (the *decoder*) to learn a distribution $q(\boldsymbol{y}|\boldsymbol{z})$ to approximate $p(\boldsymbol{y}|\boldsymbol{z})$. The approximate distribution $q(\boldsymbol{y}|\boldsymbol{z})$ serves as a substitute for $p(\boldsymbol{y}|\boldsymbol{z})$ in Eq. (9). However, we argue that directly substituting $p(\boldsymbol{y}|\boldsymbol{z})$ with $q(\boldsymbol{y}|\boldsymbol{z})$ in Eq. (9) requires a more careful examination. Specifically, expanding Eq. (9) yields:

$$\begin{aligned} I(Z, Y) &\geq \mathbb{E}_{p(\boldsymbol{x})}\mathbb{E}_{p(\boldsymbol{y}|\boldsymbol{x})}\mathbb{E}_{p(\boldsymbol{z}|\boldsymbol{x})}[\log p(\boldsymbol{y}|\boldsymbol{z})] = \mathbb{E}_{p(\boldsymbol{x})}\mathbb{E}_{p(\boldsymbol{y}|\boldsymbol{x})}\mathbb{E}_{p(\boldsymbol{z}|\boldsymbol{x})}\left[\log\left(q(\boldsymbol{y}|\boldsymbol{z})\frac{p(\boldsymbol{y}|\boldsymbol{z})}{q(\boldsymbol{y}|\boldsymbol{z})}\right)\right] \\ &= \underbrace{\mathbb{E}_{p(\boldsymbol{x})}\mathbb{E}_{p(\boldsymbol{y}|\boldsymbol{x})}\mathbb{E}_{p(\boldsymbol{z}|\boldsymbol{x})}\left[\log q(\boldsymbol{y}|\boldsymbol{z})\right]}_{\text{(i)}} + \underbrace{\mathbb{E}_{p(\boldsymbol{x})}\mathbb{E}_{p(\boldsymbol{y}|\boldsymbol{x})}\mathbb{E}_{p(\boldsymbol{z}|\boldsymbol{x})}\left[\log \frac{p(\boldsymbol{y}|\boldsymbol{z})}{q(\boldsymbol{y}|\boldsymbol{z})}\right]}_{\text{(ii)}}. \end{aligned}$$

$$\tag{11}$$

---

[1]Alemi et al. (2017) initially derive the objective using the Markov chain $Z - X - Y$ but implement it with $X - Z - Y$. Wieczorek & Roth (2020) directly derive the lower bound from $X - Z - Y$.

Term (i) is used for training the model in Alemi et al. (2017), including both the encoder and decoder. Term (ii) represents the gap between term (i) and the true lower bound in Eq. (9). Crucially, **term (ii) is not necessarily non-negative**. If term (ii) is negative, it undermines the assumption that term (i) serves as a valid lower bound on $I(Z, Y)$. This calls into question its suitability as a training objective. Term (ii) remains small only when the model is well-trained so that $q(\boldsymbol{y}|\boldsymbol{z})$ closely approximates $p(\boldsymbol{y}|\boldsymbol{z})$. In summary, when introducing IB at an intermediate layer within an LLM and substituting $p(\boldsymbol{y}|\boldsymbol{z})$ with $q(\boldsymbol{y}|\boldsymbol{z})$, we lose control over whether the training objective truly remains a lower bound on $I(Z, Y)$. In the next section, we present our strategy to address this challenge.

## 3.2 REGULARIZING EDL WITH IB

So far we have not introduced how to apply IB to the EDL pipeline: $\boldsymbol{x} \rightarrow f(\boldsymbol{x}; \boldsymbol{\theta}) \rightarrow \tilde{\boldsymbol{e}} \rightarrow \boldsymbol{e} \rightarrow \boldsymbol{\alpha} \rightarrow \boldsymbol{\pi} \rightarrow \boldsymbol{y}$. In this section, we focus on two key questions: **(1) What should the hidden variable $Z$ be in IB-EDL?** In other words, where in the EDL pipeline should we introduce IB? **(2) What prior distribution $r(\boldsymbol{z})$ should we select?**

**Choice of hidden variable $Z$:** As highlighted in Section 3.1, a key challenge in applying IB *within* a neural network is the potential violation of the lower bound. We address this by choosing the pre-evidence as the hidden variable $Z$, i.e., $\boldsymbol{z} = \tilde{\boldsymbol{e}} \in \mathbb{R}^C$. By doing so, we can *exactly evaluate* $p(\boldsymbol{y}|\boldsymbol{z})$ from the pipeline $\tilde{\boldsymbol{e}} \rightarrow \boldsymbol{e} \rightarrow \boldsymbol{\alpha} \rightarrow \boldsymbol{\pi} \rightarrow \boldsymbol{y}$. Specifically, given $\tilde{\boldsymbol{e}}$, we can compute $\boldsymbol{\alpha} = \boldsymbol{e} + \mathbf{1} = \text{SoftPlus}(\tilde{\boldsymbol{e}}) + \mathbf{1}$, from which $\boldsymbol{\pi} \sim \text{Dir}(\boldsymbol{\pi}; \boldsymbol{\alpha})$ and $\boldsymbol{y} \sim \text{Cat}(\boldsymbol{y}; \boldsymbol{\pi})$ follow. As a result, there is no need to learn $q(\boldsymbol{y}|\boldsymbol{z})$, allowing us to directly use the lower bound in Eq. (9) (and hence Eq. (10)) as the training objective, which only involves $p(\boldsymbol{y}|\boldsymbol{z})$. In other words, by choosing $\boldsymbol{z} = \tilde{\boldsymbol{e}}$, we skip the step of learning an approximated distribution $q(\boldsymbol{y}|\boldsymbol{z})$ and ensure that we are maximizing a valid lower bound of $I(Z, Y)$. Next, we proceed to compute this bound, namely the first term in Eq. (10), which can be analytically computed as:

$$\mathcal{L}_{\text{IB-NLL}}(\boldsymbol{\theta}) := -\mathbb{E}_{p(\boldsymbol{x})}\mathbb{E}_{p(\boldsymbol{y}|\boldsymbol{x})}\mathbb{E}_{p(\boldsymbol{z}|\boldsymbol{x})}[\log p(\boldsymbol{y}|\boldsymbol{z})]$$
$$= \mathbb{E}_{p(\boldsymbol{x})}\mathbb{E}_{p(\boldsymbol{y}|\boldsymbol{x})}\mathbb{E}_{p(\boldsymbol{z}|\boldsymbol{x})}\left[\sum_{j=1}^{C} y_j(\log(\alpha_0) - \log(\alpha_j))\right], \quad (12)$$

where we have omitted the dependency of $\boldsymbol{\alpha}$ on $\boldsymbol{z} = \tilde{\boldsymbol{e}}$. Furthermore, our method can be further enhanced by integrating the finding of Sensoy et al. (2018), suggesting the MSE loss as a practically more stable objective. This can be analytically computed from $\boldsymbol{\alpha}$ (and $\tilde{\boldsymbol{e}}$) as follows:

$$\mathcal{L}_{\text{IB-MSE}}(\boldsymbol{\theta}) := \mathbb{E}_{p(\boldsymbol{x})}\mathbb{E}_{p(\boldsymbol{y}|\boldsymbol{x})}\mathbb{E}_{p(\boldsymbol{z}|\boldsymbol{x})}\left[||\boldsymbol{y} - \boldsymbol{\pi}||_2^2\right]$$
$$= \mathbb{E}_{p(\boldsymbol{x})}\mathbb{E}_{p(\boldsymbol{y}|\boldsymbol{x})}\mathbb{E}_{p(\boldsymbol{z}|\boldsymbol{x})}\left[\sum_{j=1}^{C}\left(y_j - \frac{\alpha_j}{\alpha_0}\right)^2 + \frac{\alpha_j(\alpha_0 - \alpha_j)}{\alpha_0^2(\alpha_j + 1)}\right]. \quad (13)$$

**Choice of prior $r(\boldsymbol{z})$:** Having defined $\boldsymbol{z} = \tilde{\boldsymbol{e}}$, we now require a suitable prior $r(\boldsymbol{z})$ for $\boldsymbol{z}$. Notably, $\tilde{\boldsymbol{e}}$ is the output of the LLM, and recent studies (Zhang et al., 2021; Hashemi et al., 2021) suggest that activation distributions in the later layers of neural networks tend to resemble Gaussian distributions more closely than those in earlier layers. Additionally, the pre-evidence $\tilde{\boldsymbol{e}}$ represents the LLM's output values, typically ranging from -2 to 2. Therefore, a standard Gaussian prior, $z_j \sim \mathcal{N}(0, 1) \; \forall j$, represents a reasonable choice.[2] Given that $\boldsymbol{z} = \tilde{\boldsymbol{e}}$ follows a Gaussian distribution, we leverage the LLM $f$ to learn the mean and covariance of this $\boldsymbol{z}$'s distribution. Typically, an LLM comprises a sequence of transformer layers, denoted as $g$, followed by a linear head $h$, such that $f = h \circ g$. To model the Gaussian distribution, we double the number of output neurons in $h$, partitioning it into two equal-sized functional parts. One part, $h^\mu$, predicts the Gaussian mean, while the other, $h^\sigma$, predicts the variances. Since $h^\mu$ and $h^\sigma$ predictors share the same features from $g$, both predictions can be computed in a single forward pass. We define $\boldsymbol{\mu} = h^\mu(g(\boldsymbol{x}))$ and $\boldsymbol{\sigma} = \text{SoftPlus}(h^\sigma(g(\boldsymbol{x})))$, yielding $p(\boldsymbol{z}|\boldsymbol{x}; \boldsymbol{\theta}) = \mathcal{N}(\boldsymbol{z}; \boldsymbol{\mu}, \text{diag}(\boldsymbol{\sigma}))$. Then the second term in Eq. (10) becomes:

$$\mathcal{L}_{\text{IB-Info}}(\boldsymbol{\theta}) := \mathbb{E}_{p(\boldsymbol{x})}\left[D_{\text{KL}}(p(\boldsymbol{z}|\boldsymbol{x})\|r(\boldsymbol{z}))\right] \propto \frac{1}{2}\mathbb{E}_{p(\boldsymbol{x})}\left[\|\boldsymbol{\mu}\|_2^2 + \|\boldsymbol{\sigma}\|_2^2 - 2\sum_{j=1}^{C}\log(\sigma_j)\right]. \quad (14)$$

---

[2]Alternatively, we can also choose $\boldsymbol{z} = \boldsymbol{e} = \text{SoftPlus}(\tilde{\boldsymbol{e}})$; however, the prior will be a truncated Gaussian.

---

**Algorithm 1** IB-EDL training and inference pseudocode.

---

**Require:** Data $(\boldsymbol{x}, \boldsymbol{y})$, LLM $f = h \circ g$, weight $\beta$, sample size $K$, binary flag `IsTraining`.
 1: $\boldsymbol{\mu}, \boldsymbol{\sigma} \leftarrow (h \circ g)(\boldsymbol{x})$; and compute $\mathcal{L}_{\text{IB-Info}}(\boldsymbol{\theta})$ with $\boldsymbol{\mu}, \boldsymbol{\sigma}$.     ▷ *Predict $p(\tilde{e}|\boldsymbol{x})$ using the LLM.*
 2: Draw $\{\tilde{\boldsymbol{e}}^{(k)}\}_{k=1}^{K}$ from $\mathcal{N}(\tilde{\boldsymbol{e}}; \boldsymbol{\mu}, \text{diag}(\boldsymbol{\sigma}))$.     ▷ *Parallelized in PyTorch.*
 3: **if** `IsTraining` **then**     ▷ *At training time.*
 4: |   Compute $\mathcal{L}_{\text{IB-MSE}}(\boldsymbol{\theta})$ for each $\tilde{e}^{(k)}$ and take the **average**.     ▷ *Eq. (13). Also parallelized.*
 5: |   Backpropagate **averaged** $\mathcal{L}_{\text{IB-MSE}}(\boldsymbol{\theta}) + \beta \, \mathcal{L}_{\text{IB-Info}}(\boldsymbol{\theta})$.
 6: **else**     ▷ *At inference time.*
 7: |   Compute **average** $\tilde{\boldsymbol{e}} \leftarrow \frac{1}{K} \sum_{k=1}^{K} \tilde{\boldsymbol{e}}^{(k)}$; and $\boldsymbol{\alpha} \leftarrow \text{SoftPlus}(\tilde{\boldsymbol{e}}) + 1$; and $\hat{\pi}_j \leftarrow \alpha_j/\alpha_0 \; \forall j$.
 8: |   Final prediction $\hat{y} \leftarrow \arg\max_j \hat{\pi}_j$. Uncertainty mass: $u \leftarrow C/\alpha_0$.

---

**Overall IB-EDL loss and its interpretation:** The final IB-EDL objective is:

$$\min_{\boldsymbol{\theta}} \; \mathcal{L}_{\text{IB-MSE}}(\boldsymbol{\theta}) + \beta \, \mathcal{L}_{\text{IB-Info}}(\boldsymbol{\theta}). \tag{15}$$

Importantly, Eq. (14) imposes a $\ell_2$-regularization on $\boldsymbol{\mu}$, i.e. the mean of $\tilde{\boldsymbol{e}}$, and thus on $\boldsymbol{\alpha}$. *Therefore, IB-EDL penalizes the LLM for generating large $\boldsymbol{\alpha}$ that lead to over-confident predictions.*

### 3.3 Practical Implementation

In this subsection, we focus on the implementation of IB-EDL, so *we use $\tilde{e}$ instead of $\boldsymbol{z}$*. Eq. (13) requires sampling $\tilde{e}$ from the predicted distribution $\mathcal{N}(\tilde{e}; \boldsymbol{\mu}, \text{diag}(\boldsymbol{\sigma}))$ and using the sampled $\tilde{e}$ to compute $\boldsymbol{\alpha}$. To handle the non-differentiable sampling operation, we apply the reparameterization trick (Kingma & Welling, 2014), allowing gradients to flow through the LLM parameters $\boldsymbol{\theta}$. For each input $\boldsymbol{x}$, we sample $K$ (e.g. $K = 20$) pre-evidences $\tilde{e}$ and compute the average loss derived from them (see Algorithm 1). At inference time, we also sample $K$ values and compute the average $\tilde{e}$. The extra time cost is minimal compared to the inference time of the LLM (detailed in Section 4.5).

### 3.4 Variational Bayes-based EDL as a special case of IB-EDL

Some EDL methods (Chen et al., 2018; Joo et al., 2020) are based on a variational Bayes (VB) perspective and can be seen as a special case within the IB-EDL framework. More formally:

**Proposition 1.** *The VB-based EDL methods, which minimize $\mathbb{E}_{p(\boldsymbol{x}, \boldsymbol{y})} [D_{KL} (p(\boldsymbol{\pi}|\boldsymbol{x}; \boldsymbol{\theta}) || p(\boldsymbol{\pi}|\boldsymbol{y}))]$, are a special case of IB-EDL when the hidden variable is chosen as $\boldsymbol{z} = \boldsymbol{\pi}$ (i.e. token probabilities) and the prior $r(\boldsymbol{z})$ is chosen as $Dir(\boldsymbol{z}; \boldsymbol{y} \odot \boldsymbol{\alpha} + (1 - \boldsymbol{y}))$.*

The proof is detailed in the Appendix B. In the experiments, we will also compare our method with VID (Chen et al., 2018), a representative EDL method from this category.

## 4 Experiments

### 4.1 Experimental setups

**Models:** We fine-tune Llama2-7B (Touvron et al., 2023), Llama3-8B (Dubey et al., 2024), and Mistral-7B (Jiang et al., 2023) using LoRA (Hu et al., 2022) implemented via PEFT (Mangrulkar et al., 2022) and Transformers (Wolf et al., 2020). Details on training configurations and $\beta$ values are provided in Appendix C. Due to space constraints, we primarily present results for Llama2-7B and Llama3-8B, with Mistral-7B results provided in Appendix E.1.

**Datasets:** We compare methods on six multiple-choice classification datasets, including five for commonsense reasoning, ARC-C and ARC-E (Clark et al., 2018), OpenbookQA (OBQA) (Mihaylov et al., 2018), CommonsenseQA (CSQA) (Talmor et al., 2019), and SciQ (Welbl et al., 2017), alongside a dataset for reading comprehension, RACE (Lai et al., 2017). For these datasets, we define the target space $\mathcal{Y}$ as the tokens corresponding to the possible options (A/B/C/D). When fine-tuning LLMs, we select next-token logits (pre-evidences) corresponding to these options.

**Baselines:** We compare our method against a variety of baselines, including standard MAP training, two conventional uncertainty-aware approaches: Deep Ensemble (**Ens**) (Lakshminarayanan et al.,

Table 1: Calibration performance of uncertainty-aware methods on fine-tuned Llama2-7B. Arrows ("↑" or "↓") indicate whether higher or lower values signify better performance, respectively. The best and second-best results are highlighted in **bold** and underlined, respectively. Additionally, $\pm$ denotes the standard deviation of 3 runs. IB-EDL consistently achieves comparable accuracy while significantly reducing ECE and NLL, thereby greatly mitigating the overconfidence of the LLM.

| Metrics | Method | ARC-C | ARC-E | OBQA | CSQA | SciQ | RACE |
|---|---|---|---|---|---|---|---|
| Acc ↑ | MAP | $68.00_{\pm 1.72}$ | $85.07_{\pm 0.11}$ | $79.86_{\pm 0.12}$ | $78.67_{\pm 0.26}$ | $91.30_{\pm 0.17}$ | $81.29_{\pm 0.29}$ |
| | MCD | $68.01_{\pm 1.71}$ | $85.10_{\pm 0.09}$ | $79.70_{\pm 0.46}$ | $78.65_{\pm 0.30}$ | **$91.33_{\pm 0.23}$** | $81.26_{\pm 0.31}$ |
| | Ens | $67.46_{\pm 0.21}$ | $85.16_{\pm 0.25}$ | $79.72_{\pm 0.61}$ | $78.84_{\pm 0.16}$ | $90.83_{\pm 0.60}$ | $81.26_{\pm 0.18}$ |
| | LA | $66.92_{\pm 0.55}$ | $84.60_{\pm 0.44}$ | $80.15_{\pm 0.22}$ | $78.58_{\pm 0.50}$ | $90.87_{\pm 0.15}$ | $81.19_{\pm 0.33}$ |
| | EDL | $66.78_{\pm 1.69}$ | $84.56_{\pm 0.67}$ | $79.80_{\pm 0.40}$ | $79.03_{\pm 0.62}$ | $90.07_{\pm 0.61}$ | $81.03_{\pm 0.29}$ |
| | VID | $67.40_{\pm 0.90}$ | $84.69_{\pm 0.61}$ | $78.93_{\pm 0.50}$ | $78.46_{\pm 0.25}$ | $91.13_{\pm 0.42}$ | $81.33_{\pm 0.55}$ |
| | I-EDL | $66.98_{\pm 0.37}$ | $84.76_{\pm 0.69}$ | **$81.79_{\pm 0.53}$** | **$79.22_{\pm 0.39}$** | $90.43_{\pm 0.70}$ | $78.06_{\pm 0.32}$ |
| | R-EDL | **$70.37_{\pm 0.47}$** | $84.53_{\pm 0.16}$ | $81.13_{\pm 0.51}$ | $78.64_{\pm 0.16}$ | $90.49_{\pm 0.36}$ | $79.29_{\pm 1.57}$ |
| | IB-EDL | $68.17_{\pm 0.92}$ | **$85.67_{\pm 0.76}$** | $80.17_{\pm 0.20}$ | $78.62_{\pm 1.07}$ | $91.10_{\pm 0.66}$ | **$81.82_{\pm 0.30}$** |
| ECE ↓ | MAP | $30.08_{\pm 0.96}$ | $13.66_{\pm 0.14}$ | $17.81_{\pm 0.14}$ | $19.06_{\pm 0.25}$ | $6.85_{\pm 0.21}$ | $9.62_{\pm 0.32}$ |
| | MCD | $30.08_{\pm 0.97}$ | $12.87_{\pm 0.71}$ | $17.14_{\pm 0.77}$ | $19.06_{\pm 0.25}$ | $6.83_{\pm 0.20}$ | $9.62_{\pm 0.32}$ |
| | Ens | $28.26_{\pm 2.78}$ | $12.64_{\pm 0.72}$ | $15.19_{\pm 0.94}$ | $17.82_{\pm 0.17}$ | $6.77_{\pm 0.33}$ | $9.51_{\pm 0.43}$ |
| | LA | $8.80_{\pm 3.77}$ | $25.01_{\pm 1.32}$ | $10.71_{\pm 0.16}$ | $11.81_{\pm 1.56}$ | $15.51_{\pm 2.79}$ | $9.62_{\pm 0.19}$ |
| | EDL | $14.26_{\pm 1.38}$ | $8.87_{\pm 1.40}$ | $9.14_{\pm 1.98}$ | $9.71_{\pm 1.04}$ | $18.69_{\pm 1.17}$ | $7.47_{\pm 0.72}$ |
| | VID | $13.94_{\pm 0.87}$ | $4.78_{\pm 0.23}$ | $10.06_{\pm 1.51}$ | $8.25_{\pm 0.27}$ | $6.26_{\pm 0.54}$ | $4.56_{\pm 0.81}$ |
| | I-EDL | $12.39_{\pm 0.16}$ | $11.33_{\pm 0.27}$ | $10.84_{\pm 1.18}$ | $11.66_{\pm 0.61}$ | $15.13_{\pm 0.08}$ | $13.52_{\pm 0.85}$ |
| | R-EDL | $15.34_{\pm 2.81}$ | $4.68_{\pm 0.41}$ | $7.10_{\pm 0.68}$ | $5.77_{\pm 0.66}$ | $6.13_{\pm 0.25}$ | $4.48_{\pm 0.59}$ |
| | IB-EDL | **$6.38_{\pm 0.56}$** | **$2.57_{\pm 0.57}$** | **$5.84_{\pm 0.88}$** | **$4.90_{\pm 1.08}$** | **$5.01_{\pm 0.43}$** | **$4.03_{\pm 0.18}$** |
| NLL ↓ | MAP | $2.98_{\pm 0.11}$ | $1.09_{\pm 0.04}$ | $1.14_{\pm 0.03}$ | $1.26_{\pm 0.01}$ | $0.38_{\pm 0.01}$ | $0.61_{\pm 0.01}$ |
| | MCD | $2.98_{\pm 0.11}$ | $1.09_{\pm 0.05}$ | $1.11_{\pm 0.04}$ | $1.26_{\pm 0.01}$ | $0.38_{\pm 0.01}$ | $0.61_{\pm 0.01}$ |
| | Ens | $2.90_{\pm 0.19}$ | $1.08_{\pm 0.03}$ | $1.05_{\pm 0.05}$ | $1.12_{\pm 0.06}$ | $0.39_{\pm 0.02}$ | $0.59_{\pm 0.01}$ |
| | LA | **$1.03_{\pm 0.01}$** | $0.70_{\pm 0.01}$ | **$0.61_{\pm 0.00}$** | $0.70_{\pm 0.03}$ | $0.37_{\pm 0.03}$ | **$0.56_{\pm 0.01}$** |
| | EDL | $1.07_{\pm 0.04}$ | $0.59_{\pm 0.03}$ | $0.65_{\pm 0.01}$ | $0.75_{\pm 0.01}$ | $0.47_{\pm 0.01}$ | $0.62_{\pm 0.01}$ |
| | VID | $1.07_{\pm 0.01}$ | $0.60_{\pm 0.02}$ | $0.71_{\pm 0.02}$ | $0.77_{\pm 0.01}$ | $0.31_{\pm 0.01}$ | $0.63_{\pm 0.03}$ |
| | I-EDL | $1.08_{\pm 0.01}$ | $0.60_{\pm 0.02}$ | $0.62_{\pm 0.01}$ | $0.75_{\pm 0.01}$ | $0.46_{\pm 0.01}$ | $0.71_{\pm 0.01}$ |
| | R-EDL | $1.07_{\pm 0.05}$ | $0.57_{\pm 0.01}$ | **$0.61_{\pm 0.01}$** | $0.74_{\pm 0.01}$ | $0.35_{\pm 0.01}$ | $0.61_{\pm 0.03}$ |
| | IB-EDL | **$1.03_{\pm 0.02}$** | **$0.53_{\pm 0.03}$** | $0.65_{\pm 0.01}$ | **$0.74_{\pm 0.02}$** | **$0.29_{\pm 0.01}$** | **$0.50_{\pm 0.01}$** |

2017; Fort et al., 2019) and MC-Dropout (**MCD**) (Gal & Ghahramani, 2016), and Laplace-LoRA (**LA**) (Yang et al., 2024), a recent calibration method tailored for fine-tuned LLMs. Additionally, we include four baselines from the EDL family: vanilla **EDL** (Sensoy et al., 2018), **VID** (Chen et al., 2018) from VB-based EDL, and two SOTA methods: **I-EDL** (Deng et al., 2023) and **R-EDL** (Chen et al., 2024). We use their original implementations and hyperparameters where available. Additionally, although PostNet (Charpentier et al., 2020) is also a well-known EDL method, it is omitted here because it requires a specialized Normalizing Flow design to be compatible with Transformers.

## 4.2 IN-DISTRIBUTION CALIBRATION

An effective uncertainty-aware method should 1) significantly improve model calibration and 2) show accuracy comparable to standard MAP training. We therefore use Accuracy (Acc), expected calibration error (ECE), and negative log-likelihood (NLL) as metrics for evaluating the fine-tuned LLMs on the six aforementioned datasets. Table 1 and Table 2 present the results for Llama2-7B and Llama3-8B, respectively. The accuracy of IB-EDL and other uncertainty-aware methods is on par with, or even higher than, the MAP baseline, so we focus primarily on analyzing ECE and NLL. MAP exhibits substantially higher ECE and NLL than other methods, suggesting that fine-tuning LLMs on small datasets using MAP (or MLE) leads to significant overconfidence. Overall, IB-EDL shows the lowest ECE and NLL, reducing ECE by several factors compared to MAP, MCD, and Ens. This highlights IB-EDL's ability to effectively mitigate overconfidence in fine-tuned models. The superior performance of IB-EDL, compared to other EDL methods, can be attributed to the $\ell_2$ regularization in Eq. (14), which discourages the model from generating excessively large evidences that lead to over-concentrated Dirichlet distributions. Furthermore, the advantages of IB-EDL extend to other architectures, such as Mistral-7B, as demonstrated in Appendix E.1.

Table 2: Calibration performance of uncertainty-aware methods on fine-tuned Llama3-8B.

| Metrics | Method | ARC-C | ARC-E | OBQA | CSQA | SciQ | RACE |
|---|---|---|---|---|---|---|---|
| Acc ↑ | MAP | $79.74_{\pm0.27}$ | $92.27_{\pm0.17}$ | $88.60_{\pm0.87}$ | $81.51_{\pm0.60}$ | $93.37_{\pm0.12}$ | $\underline{88.15}_{\pm0.31}$ |
|  | MCD | $79.55_{\pm0.21}$ | $92.25_{\pm0.13}$ | $\underline{88.63}_{\pm0.84}$ | $\underline{81.52}_{\pm0.60}$ | $93.30_{\pm0.20}$ | $88.12_{\pm0.25}$ |
|  | Ens | $79.50_{\pm0.10}$ | $92.26_{\pm0.41}$ | $88.53_{\pm0.59}$ | $81.39_{\pm0.63}$ | $93.27_{\pm0.15}$ | $88.09_{\pm0.08}$ |
|  | LA | $77.79_{\pm0.39}$ | $92.18_{\pm0.24}$ | $88.34_{\pm0.62}$ | $81.30_{\pm0.44}$ | $93.37_{\pm0.15}$ | $88.09_{\pm0.19}$ |
|  | EDL | $79.35_{\pm1.11}$ | $92.31_{\pm0.76}$ | $87.67_{\pm0.31}$ | $80.67_{\pm0.29}$ | $93.13_{\pm0.36}$ | $87.76_{\pm0.21}$ |
|  | VID | $79.99_{\pm0.13}$ | $92.30_{\pm0.25}$ | $87.57_{\pm0.29}$ | $80.82_{\pm0.94}$ | $92.93_{\pm0.23}$ | $\mathbf{88.23}_{\pm0.25}$ |
|  | I-EDL | $80.37_{\pm0.48}$ | $\mathbf{92.76}_{\pm0.47}$ | $88.52_{\pm0.50}$ | $81.13_{\pm0.39}$ | $\underline{93.47}_{\pm0.06}$ | $85.91_{\pm0.40}$ |
|  | R-EDL | $\underline{80.60}_{\pm0.93}$ | $92.41_{\pm0.30}$ | $88.00_{\pm0.20}$ | $80.83_{\pm1.00}$ | $93.33_{\pm0.21}$ | $87.62_{\pm0.16}$ |
|  | IB-EDL | $\mathbf{81.14}_{\pm0.09}$ | $\underline{92.55}_{\pm0.15}$ | $\mathbf{89.00}_{\pm0.40}$ | $\mathbf{81.71}_{\pm0.38}$ | $\mathbf{93.57}_{\pm0.15}$ | $88.03_{\pm0.21}$ |
| ECE ↓ | MAP | $19.68_{\pm0.43}$ | $7.18_{\pm0.14}$ | $10.52_{\pm0.87}$ | $17.29_{\pm0.57}$ | $5.74_{\pm0.08}$ | $7.95_{\pm0.35}$ |
|  | MCD | $19.91_{\pm0.39}$ | $7.10_{\pm0.02}$ | $10.48_{\pm0.86}$ | $16.98_{\pm0.10}$ | $5.74_{\pm0.09}$ | $7.93_{\pm0.35}$ |
|  | Ens | $18.20_{\pm0.17}$ | $3.81_{\pm1.60}$ | $10.08_{\pm0.90}$ | $16.11_{\pm1.63}$ | $5.72_{\pm0.24}$ | $7.80_{\pm0.19}$ |
|  | LA | $18.49_{\pm0.44}$ | $3.33_{\pm0.66}$ | $5.26_{\pm1.30}$ | $6.62_{\pm0.10}$ | $\mathbf{2.47}_{\pm0.08}$ | $\underline{3.61}_{\pm0.27}$ |
|  | EDL | $6.52_{\pm0.12}$ | $5.94_{\pm0.87}$ | $8.28_{\pm1.62}$ | $7.43_{\pm1.48}$ | $11.13_{\pm1.11}$ | $6.51_{\pm0.80}$ |
|  | VID | $10.96_{\pm0.40}$ | $3.33_{\pm1.40}$ | $5.99_{\pm1.41}$ | $8.38_{\pm0.93}$ | $\underline{2.60}_{\pm0.13}$ | $4.58_{\pm0.14}$ |
|  | I-EDL | $\underline{5.08}_{\pm1.94}$ | $9.69_{\pm0.58}$ | $7.57_{\pm0.52}$ | $8.95_{\pm0.59}$ | $13.07_{\pm0.24}$ | $14.52_{\pm1.15}$ |
|  | R-EDL | $10.09_{\pm1.01}$ | $\underline{2.93}_{\pm1.32}$ | $\underline{4.68}_{\pm1.35}$ | $\underline{6.59}_{\pm0.78}$ | $3.18_{\pm0.18}$ | $\mathbf{2.61}_{\pm0.09}$ |
|  | IB-EDL | $\mathbf{2.78}_{\pm0.87}$ | $\mathbf{2.70}_{\pm0.58}$ | $\mathbf{2.34}_{\pm0.61}$ | $\mathbf{4.34}_{\pm0.20}$ | $3.86_{\pm0.86}$ | $4.47_{\pm0.31}$ |
| NLL ↓ | MAP | $2.25_{\pm0.08}$ | $0.61_{\pm0.02}$ | $0.78_{\pm0.05}$ | $1.25_{\pm0.04}$ | $0.36_{\pm0.00}$ | $0.47_{\pm0.01}$ |
|  | MCD | $2.27_{\pm0.09}$ | $0.59_{\pm0.00}$ | $0.77_{\pm0.05}$ | $1.22_{\pm0.03}$ | $0.36_{\pm0.01}$ | $0.45_{\pm0.03}$ |
|  | Ens | $2.02_{\pm0.07}$ | $0.43_{\pm0.09}$ | $0.74_{\pm0.05}$ | $1.22_{\pm0.12}$ | $0.35_{\pm0.01}$ | $0.46_{\pm0.05}$ |
|  | LA | $0.80_{\pm0.01}$ | $0.33_{\pm0.04}$ | $\underline{0.42}_{\pm0.01}$ | $\mathbf{0.62}_{\pm0.04}$ | $\mathbf{0.22}_{\pm0.00}$ | $0.37_{\pm0.01}$ |
|  | EDL | $0.74_{\pm0.02}$ | $0.33_{\pm0.01}$ | $0.45_{\pm0.02}$ | $0.68_{\pm0.01}$ | $0.31_{\pm0.01}$ | $0.43_{\pm0.01}$ |
|  | VID | $0.78_{\pm0.01}$ | $0.35_{\pm0.01}$ | $0.46_{\pm0.02}$ | $0.72_{\pm0.03}$ | $0.28_{\pm0.01}$ | $\underline{0.36}_{\pm0.00}$ |
|  | I-EDL | $\underline{0.72}_{\pm0.02}$ | $0.36_{\pm0.02}$ | $0.43_{\pm0.00}$ | $0.68_{\pm0.01}$ | $0.32_{\pm0.01}$ | $0.52_{\pm0.02}$ |
|  | R-EDL | $0.74_{\pm0.01}$ | $\mathbf{0.32}_{\pm0.01}$ | $0.43_{\pm0.02}$ | $0.68_{\pm0.02}$ | $0.27_{\pm0.01}$ | $0.43_{\pm0.02}$ |
|  | IB-EDL | $\mathbf{0.69}_{\pm0.01}$ | $\mathbf{0.32}_{\pm0.02}$ | $\mathbf{0.40}_{\pm0.03}$ | $\underline{0.66}_{\pm0.01}$ | $\underline{0.25}_{\pm0.01}$ | $\mathbf{0.35}_{\pm0.00}$ |

Table 3: OOD detection performance on fine-tuned Llama2-7B and Llama3-8B. $A \rightarrow B$ indicates $A$ as the ID training set and $B$ as the OOD test set. MP and UM are two scores for measuring AUROC.

| Model | Method | OBQA → ARC-C | | OBQA → ARC-E | | OBQA → CSQA | |
|---|---|---|---|---|---|---|---|
|  |  | AUROC ↑ | | AUROC ↑ | | AUROC ↑ | |
|  |  | MP | UM | MP | UM | MP | UM |
| Llama2-7B | MAP | $76.07_{\pm0.71}$ | – | $72.21_{\pm0.60}$ | – | $72.45_{\pm0.20}$ | – |
|  | MCD | $76.07_{\pm0.71}$ | – | $72.20_{\pm0.60}$ | – | $72.49_{\pm0.17}$ | – |
|  | Ens | $71.89_{\pm2.79}$ | – | $70.30_{\pm0.43}$ | – | $70.50_{\pm1.12}$ | – |
|  | LA | $72.85_{\pm0.86}$ | – | $67.46_{\pm0.87}$ | – | $70.71_{\pm0.40}$ | – |
|  | EDL | $68.42_{\pm1.72}$ | $74.89_{\pm1.12}$ | $78.34_{\pm0.45}$ | $74.75_{\pm0.59}$ | $76.46_{\pm1.71}$ | $72.56_{\pm1.99}$ |
|  | VID | $\underline{86.47}_{\pm0.26}$ | $\underline{81.24}_{\pm0.81}$ | $\underline{86.18}_{\pm0.79}$ | $\underline{83.14}_{\pm1.54}$ | $\underline{85.41}_{\pm0.78}$ | $\underline{77.27}_{\pm2.06}$ |
|  | I-EDL | $81.34_{\pm1.41}$ | $75.78_{\pm0.94}$ | $77.22_{\pm0.75}$ | $72.10_{\pm1.74}$ | $75.78_{\pm0.75}$ | $71.60_{\pm1.11}$ |
|  | R-EDL | $76.66_{\pm1.07}$ | $73.80_{\pm0.74}$ | $72.01_{\pm1.28}$ | $69.02_{\pm0.91}$ | $71.01_{\pm1.47}$ | $68.22_{\pm1.43}$ |
|  | IB-EDL | $\mathbf{87.47}_{\pm0.91}$ | $\mathbf{88.34}_{\pm3.07}$ | $\mathbf{86.42}_{\pm0.87}$ | $\mathbf{88.52}_{\pm2.90}$ | $\mathbf{86.38}_{\pm0.53}$ | $\mathbf{79.79}_{\pm4.64}$ |
| Llama3-8B | MAP | $63.12_{\pm1.21}$ | – | $58.11_{\pm1.55}$ | – | $64.43_{\pm1.63}$ | – |
|  | MCD | $62.81_{\pm1.00}$ | – | $58.30_{\pm1.98}$ | – | $64.05_{\pm2.09}$ | – |
|  | Ens | $62.70_{\pm1.04}$ | – | $58.19_{\pm1.72}$ | – | $63.79_{\pm0.82}$ | – |
|  | LA | $62.14_{\pm0.55}$ | – | $56.11_{\pm0.84}$ | – | $63.25_{\pm1.06}$ | – |
|  | EDL | $82.04_{\pm0.69}$ | $78.99_{\pm1.18}$ | $78.80_{\pm1.47}$ | $74.77_{\pm2.16}$ | $81.07_{\pm1.22}$ | $77.55_{\pm1.69}$ |
|  | VID | $\underline{88.85}_{\pm1.57}$ | $\underline{89.95}_{\pm1.59}$ | $\underline{87.55}_{\pm0.94}$ | $\underline{90.02}_{\pm1.72}$ | $\underline{88.66}_{\pm1.75}$ | $\underline{84.37}_{\pm0.20}$ |
|  | I-EDL | $81.31_{\pm0.52}$ | $78.54_{\pm0.61}$ | $78.29_{\pm1.14}$ | $74.79_{\pm1.43}$ | $77.85_{\pm3.29}$ | $73.80_{\pm4.23}$ |
|  | R-EDL | $75.79_{\pm0.51}$ | $73.64_{\pm0.49}$ | $71.51_{\pm0.79}$ | $68.81_{\pm0.79}$ | $70.60_{\pm0.82}$ | $67.42_{\pm1.30}$ |
|  | IB-EDL | $\mathbf{88.85}_{\pm0.96}$ | $\mathbf{92.58}_{\pm0.37}$ | $\mathbf{88.14}_{\pm1.10}$ | $\mathbf{94.77}_{\pm0.42}$ | $\mathbf{89.16}_{\pm1.01}$ | $\mathbf{85.45}_{\pm0.55}$ |

## 4.3 OUT-OF-DISTRIBUTION DETECTION

In addition to in-distribution (ID) calibration, out-of-distribution (OOD) detection serves as a key benchmark for assessing the performance of uncertainty-aware methods. An effective approach should reliably assign higher uncertainties to OOD samples compared to ID samples. This can

Table 4: Fine-tuning Llama2-7B and Llama3-8B on noisy datasets. In each training dataset, the labels of 30% samples are randomly perturbed. IB-EDL is more robust to noise than other baselines.

| Metric | Method | ARC-C | ARC-E | OBQA | CSQA | SciQ | RACE |
|---|---|---|---|---|---|---|---|
| Llama2-7B Acc ↑ | MAP | $51.08_{\pm 2.96}$ | $71.39_{\pm 2.73}$ | $73.93_{\pm 2.21}$ | $74.56_{\pm 0.31}$ | $88.63_{\pm 0.06}$ | $76.83_{\pm 0.39}$ |
| | MCD | $51.08_{\pm 2.96}$ | $71.40_{\pm 2.72}$ | $73.93_{\pm 2.20}$ | $74.56_{\pm 0.32}$ | $88.64_{\pm 0.06}$ | $76.95_{\pm 0.19}$ |
| | Ens | $56.10_{\pm 3.11}$ | $76.03_{\pm 1.99}$ | $75.23_{\pm 1.06}$ | $74.59_{\pm 0.23}$ | $88.64_{\pm 0.06}$ | $76.94_{\pm 0.22}$ |
| | LA | $53.38_{\pm 2.17}$ | $74.90_{\pm 1.76}$ | $74.57_{\pm 2.22}$ | $74.59_{\pm 0.20}$ | $88.47_{\pm 0.21}$ | $77.04_{\pm 0.39}$ |
| | EDL | $57.16_{\pm 3.33}$ | $76.16_{\pm 1.12}$ | $74.46_{\pm 1.33}$ | $74.65_{\pm 0.05}$ | $89.03_{\pm 0.25}$ | $77.69_{\pm 0.44}$ |
| | VID | $\underline{57.56}_{\pm 0.58}$ | $76.57_{\pm 1.79}$ | $76.67_{\pm 1.42}$ | $\underline{75.97}_{\pm 0.66}$ | $89.06_{\pm 0.21}$ | $\underline{78.75}_{\pm 0.25}$ |
| | I-EDL | $53.86_{\pm 1.25}$ | $76.08_{\pm 0.71}$ | $\underline{77.00}_{\pm 0.88}$ | $74.01_{\pm 1.28}$ | $\mathbf{89.26}_{\pm 0.32}$ | $76.48_{\pm 0.87}$ |
| | R-EDL | $57.00_{\pm 0.76}$ | $\underline{76.96}_{\pm 1.07}$ | $73.20_{\pm 1.44}$ | $74.36_{\pm 0.49}$ | $87.33_{\pm 0.49}$ | $78.01_{\pm 0.53}$ |
| | IB-EDL | $\mathbf{59.06}_{\pm 2.07}$ | $\mathbf{80.27}_{\pm 0.48}$ | $\mathbf{78.13}_{\pm 0.99}$ | $\mathbf{76.35}_{\pm 0.66}$ | $89.06_{\pm 0.50}$ | $\mathbf{79.41}_{\pm 0.59}$ |
| Llama3-8B Acc ↑ | MAP | $57.71_{\pm 0.42}$ | $80.34_{\pm 1.47}$ | $78.78_{\pm 1.00}$ | $77.04_{\pm 0.41}$ | $92.60_{\pm 0.53}$ | $86.93_{\pm 0.11}$ |
| | MCD | $57.71_{\pm 0.42}$ | $80.37_{\pm 1.46}$ | $79.00_{\pm 0.92}$ | $77.05_{\pm 0.41}$ | $92.87_{\pm 0.12}$ | $86.93_{\pm 0.12}$ |
| | Ens | $63.39_{\pm 1.09}$ | $84.63_{\pm 0.53}$ | $80.61_{\pm 0.53}$ | $77.62_{\pm 0.41}$ | $92.97_{\pm 0.06}$ | $86.93_{\pm 0.07}$ |
| | LA | $66.62_{\pm 1.08}$ | $82.64_{\pm 0.50}$ | $79.59_{\pm 0.72}$ | $77.80_{\pm 0.49}$ | $92.93_{\pm 0.21}$ | $\underline{87.00}_{\pm 0.01}$ |
| | EDL | $\underline{69.43}_{\pm 0.98}$ | $87.57_{\pm 0.13}$ | $85.60_{\pm 0.72}$ | $79.14_{\pm 0.41}$ | $92.90_{\pm 0.40}$ | $86.26_{\pm 0.76}$ |
| | VID | $66.58_{\pm 1.92}$ | $\underline{87.67}_{\pm 0.99}$ | $84.86_{\pm 1.01}$ | $\mathbf{79.66}_{\pm 1.26}$ | $93.23_{\pm 0.25}$ | $86.86_{\pm 0.26}$ |
| | I-EDL | $63.87_{\pm 2.65}$ | $84.06_{\pm 2.80}$ | $84.26_{\pm 0.42}$ | $78.02_{\pm 0.87}$ | $92.53_{\pm 0.37}$ | $86.01_{\pm 0.51}$ |
| | R-EDL | $\mathbf{71.25}_{\pm 1.20}$ | $84.91_{\pm 3.91}$ | $\underline{85.73}_{\pm 0.70}$ | $78.57_{\pm 0.21}$ | $\mathbf{93.56}_{\pm 0.05}$ | $86.16_{\pm 0.42}$ |
| | IB-EDL | $68.53_{\pm 0.25}$ | $\mathbf{88.05}_{\pm 0.43}$ | $\mathbf{86.13}_{\pm 0.51}$ | $\underline{79.59}_{\pm 0.79}$ | $\underline{93.46}_{\pm 0.38}$ | $\mathbf{87.01}_{\pm 0.20}$ |

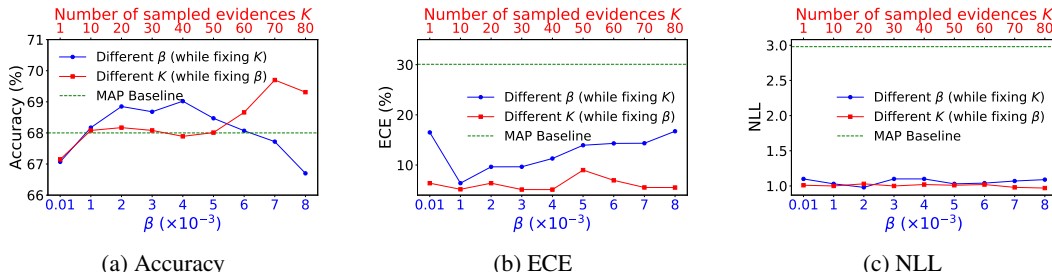

(a) Accuracy          (b) ECE          (c) NLL

Figure 1: Ablation study. IB-EDL reduces ECE and NLL compared to MAP across a broad range of $\beta$ and $K$ values. $\beta$ controls the regularization strength and balances the calibration and accuracy.

be evaluated by labeling ID samples as class 1 and OOD samples as class 0, and measuring the **AUROC** based on OOD detection scores derived from the fine-tuned model. A higher AUROC indicates better OOD detection performance. Similar to Chen et al. (2024), we use two OOD detection scores: *max probability* (**MP**) and the *reciprocal of uncertainty mass* (**UM**). We fine-tune the LLMs on OBQA (as the ID dataset) and test them on ARC-C, ARC-E, and CSQA (as OOD dataset). Note that non-EDL methods, such as LA, do not provide UM, so we evaluate them using MP only. As shown in Table 3, IB-EDL achieves the highest AUROC across all datasets using both scores, surpassing both non-EDL and EDL competitors. Furthermore, IB-EDL also demonstrates superior OOD detection performance under large distribution shifts (see Appendix E.3). Its calibration performance also generalizes well to OOD datasets (see Appendix E.4).

## 4.4 FINE-TUNING WITH NOISY LABELS

Datasets used for fine-tuning LLMs can contain a significant portion of mislabeled samples (Wang et al., 2024b; Havrilla & Iyer, 2024), and label verification is often expert-knowledge demanding. Therefore, it is crucial for fine-tuning algorithms to be robust to label noise (Wang et al., 2023a). To assess this robustness, we perturb the A/B/C/D options for 30% of the samples in each training set, fine-tune the models using the aforementioned methods on the noisy datasets, and evaluate them on clean test sets. In this adversarial setting, the primary goal is to maintain accuracy despite label noise, so we primarily evaluate accuracy, with additional metrics in the Appendix E.2. As shown in Table 4, IB-EDL achieves the highest accuracy overall, demonstrating strong robustness to label noise. This suggests that the information bottleneck effectively filters out spurious signals and retains predictive information in the generated evidence.

### 4.5 ABLATION STUDY

**Hyperparameters:** We study the weight $\beta$ and sample size $K$ (used for drawing $\tilde{e}$), using ID calibration for Llama2-7B on ARC-C as the evaluation task. Fig. 1 shows that IB-EDL consistently outperforms MAP across a broad range of values, demonstrating its robustness in reducing overconfidence. While increasing $K$ slightly improves accuracy, it does not necessarily improve ECE or NLL. $\beta$ plays a key role in balancing regularization and predictive performance. Nonetheless, all $\beta$ values reduce ECE by at least $40\%$ and NLL by at least $50\%$ compared to MAP. Additionally, we also present a sensitivity analysis on the number bins of ECE in Appendix E.6.

**Complexity analysis:** Here, we consider Llama2-7B and assume the target space is the vocabulary $\mathcal{Y} = \mathcal{V}$. Compared to the pretrained LLM, IB-EDL introduces an additional linear head $h^\sigma$, adding only $1.95\%$ more parameters. The computational overhead stems from $h^\sigma$ and the evidence averaging operation (see Algorithm 1), which amount to only $1.98\%$ of the pretrained model's GFLOPs. In Appendix E.5, we provide detailed tests of training and inference time as well as memory usage.

## 5 RELATED WORK

**EDL:** EDL leverages models to predict the Dirichlet prior distribution, with training typically done using NLL (Haussmann et al., 2023), $\ell_p$-loss (Tsiligkaridis, 2021), or MSE (Sensoy et al., 2018). Chen et al. (2018); Joo et al. (2020); Shen et al. (2023) derive loss functions from a variational Bayesian perspective, while Posterior Networks (Charpentier et al., 2020; 2022) optimize the posterior via Normalizing Flows. EDL methods often incorporate two main types of regularization: (i) encouraging uniformity in non-target Dirichlet parameters (Malinin & Gales, 2018; Sensoy et al., 2018; Chen et al., 2018; Tsiligkaridis, 2021), or (ii) modifying assumptions in the EDL formulation (Deng et al., 2023; Chen et al., 2024). IB-EDL differs by not requiring (i) and taking an alternative approach to (ii), without altering EDL assumptions. Some EDL methods also incorporate OOD data during training (Malinin & Gales, 2018; 2019). Beyond classification, EDL has been extended to regression (Amini et al., 2020) and other applications (Gao et al., 2024; Liu & Ji, 2024). Recent works (Shen et al., 2024; Juergens et al., 2024) analyze EDL's effectiveness and limitations.

**Other uncertainty-aware methods:** Besides EDL, there are other methods for uncertainty estimation and calibration, including Bayesian Neural Networks via Variational Inference (Graves, 2011; Blundell et al., 2015), MC-Dropout (Gal & Ghahramani, 2016), stochastic gradient MCMC (Welling & Teh, 2011; Ma et al., 2015), and Laplace approximations (Ritter et al., 2018; Kristiadi et al., 2021), recently extended to LoRA fine-tuned LLMs (Yang et al., 2024; Wang et al., 2024a; Li et al., 2024).

**IB:** The Information Bottleneck (IB) was introduced by Tishby et al. (1999) and later applied in neural networks for learning generalized representations (Tishby & Zaslavsky, 2015; Alemi et al., 2017; Sun et al., 2022), and as a feature attribution method (Schulz et al., 2020; Zhang et al., 2021; Wang et al., 2023b). Other works focusing on theory studied different Markov chains in IB (Wieczorek & Roth, 2020) and the impact of IB on generalization errors (Kawaguchi et al., 2023).

## 6 CONCLUSION

**Summary:** We focused on a key challenge in fine-tuning LLMs: mitigating overconfidence and improving calibration. We introduced an information-theoretic regularization to conventional EDL to prevent over-concentrated distributions in predictions. Our method, IB-EDL, introduces minimal computational overhead while significantly improving calibration in fine-tuned LLMs. Additionally, IB-EDL maintains model performance even in the presence of substantial label noise. These results highlight IB-EDL as a promising method for fostering more trustworthy LLMs.

**Limitations and future work:** Despite IB-EDL's efficiency, there is room for improvement. To reduce the complexity of covariance matrix prediction, we assume the pre-evidences are uncorrelated, but this assumption can be relaxed. Additionally, our evaluations primarily focused on conventional classification tasks, where well-established metrics for calibration and uncertainty estimation are available. In future work, it would be interesting to test IB-EDL on generative tasks. A great challenge is that uncertainty estimation metrics for generative tasks are still an ongoing research topic (Yadkori et al., 2024; Jesson et al., 2024).

ACKNOWLEDGMENT

This work is supported by the Munich Center for Machine Learning (MCML). In addition, we sincerely appreciate the insightful discussions with Emanuel Sommer and Jianfei Li.

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

## A  DERIVATION OF THE VARIATIONAL BOUNDS

**Upper bound of** $I(Z, X)$**:** We reproduce the derivation steps from Alemi et al. (2017) as follows:

$$I(Z, X) = \int p(\boldsymbol{x}, \boldsymbol{z}) \log \frac{p(\boldsymbol{z}|\boldsymbol{x})}{p(\boldsymbol{z})} \mathrm{d}\boldsymbol{z}\mathrm{d}\boldsymbol{x} = \mathbb{E}_{p(\boldsymbol{z}|\boldsymbol{x})}\mathbb{E}_{p(\boldsymbol{x})}[\log p(\boldsymbol{z}|\boldsymbol{x})] - \mathbb{E}_{p(\boldsymbol{z})}[\log p(\boldsymbol{z})]. \tag{16}$$

Given a prior $r(\boldsymbol{z})$, we have $D_{\mathrm{KL}}\left(p(\boldsymbol{z})\|r(\boldsymbol{z})\right) \geq 0$. This indicates that:

$$\begin{aligned} D_{\mathrm{KL}}\left(p(\boldsymbol{z})\|r(\boldsymbol{z})\right) &= \int p(\boldsymbol{z}) \log \frac{p(\boldsymbol{z})}{r(\boldsymbol{z})} \mathrm{d}\boldsymbol{z} \\ &= \mathbb{E}_{p(\boldsymbol{z})}[\log p(\boldsymbol{z})] - \mathbb{E}_{p(\boldsymbol{z})}[\log r(\boldsymbol{z})] \geq 0. \end{aligned}$$

Therefore,

$$\mathbb{E}_{p(\boldsymbol{z})}[\log p(\boldsymbol{z})] \geq \mathbb{E}_{p(\boldsymbol{z})}[\log r(\boldsymbol{z})]. \tag{17}$$

Plugging Eq. (17) into Eq. (16), we obtain

$$\begin{aligned} I(Z, X) &= \mathbb{E}_{p(\boldsymbol{z}|\boldsymbol{x})}\mathbb{E}_{p(\boldsymbol{x})}[\log p(\boldsymbol{z}|\boldsymbol{x})] - \mathbb{E}_{p(\boldsymbol{z})}[\log p(\boldsymbol{z})] \\ &\leq \mathbb{E}_{p(\boldsymbol{z}|\boldsymbol{x})}\mathbb{E}_{p(\boldsymbol{x})}[\log p(\boldsymbol{z}|\boldsymbol{x})] - \mathbb{E}_{p(\boldsymbol{z})}[\log r(\boldsymbol{z})] \\ &\leq \mathbb{E}_{p(\boldsymbol{z}|\boldsymbol{x})}\mathbb{E}_{p(\boldsymbol{x})}[\log p(\boldsymbol{z}|\boldsymbol{x})] - \mathbb{E}_{p(\boldsymbol{x})}\mathbb{E}_{p(\boldsymbol{z}|\boldsymbol{x})}[\log r(\boldsymbol{z})] \\ &= \mathbb{E}_{p(\boldsymbol{x})}\mathbb{E}_{p(\boldsymbol{z}|\boldsymbol{x})}\left[\log \frac{p(\boldsymbol{z}|\boldsymbol{x})}{r(\boldsymbol{z})}\right] \\ &= \mathbb{E}_{p(\boldsymbol{x})}\left[D_{\mathrm{KL}}\left(p(\boldsymbol{z}|\boldsymbol{x})\|r(\boldsymbol{z})\right)\right]. \end{aligned} \tag{18}$$

**Lower bound of** $I(Z, Y)$**:** For clarity, we reproduce the derivation steps from Wieczorek & Roth (2020) here. Unlike Alemi et al. (2017), who assume the Markov chain $Z - X - Y$, Wieczorek & Roth (2020) assume the Markov chain $X - Z - Y$, implying the conditional independence $p(\boldsymbol{y}|\boldsymbol{z}) = p(\boldsymbol{y}|\boldsymbol{z}, \boldsymbol{x})$. The detailed steps are as follows:

$$\begin{aligned} I(Z, Y) &= \mathbb{E}_{p(\boldsymbol{x}, \boldsymbol{y})}\mathbb{E}_{p(\boldsymbol{z}|\boldsymbol{x}, \boldsymbol{y})}[\log p(\boldsymbol{y}|\boldsymbol{z})] + H(Y) \\ &= \mathbb{E}_{p(\boldsymbol{x})}\mathbb{E}_{p(\boldsymbol{y}|\boldsymbol{x})}\mathbb{E}_{p(\boldsymbol{z}|\boldsymbol{x}, \boldsymbol{y})}[\log p(\boldsymbol{y}|\boldsymbol{z}, \boldsymbol{x})] + H(Y) \end{aligned} \tag{19}$$

Now, we derive a lower bound on the term $\mathbb{E}_{p(\boldsymbol{y}|\boldsymbol{x})}\mathbb{E}_{p(\boldsymbol{z}|\boldsymbol{x}, \boldsymbol{y})}[\log p(\boldsymbol{y}|\boldsymbol{z}, \boldsymbol{x})]$ in Eq. (19) as follows:

$$\begin{aligned} &\mathbb{E}_{p(\boldsymbol{y}|\boldsymbol{x})}\mathbb{E}_{p(\boldsymbol{z}|\boldsymbol{x}, \boldsymbol{y})}[\log p(\boldsymbol{y}|\boldsymbol{z}, \boldsymbol{x})] \\ &= \int\int p(\boldsymbol{z}, \boldsymbol{y}|\boldsymbol{x}) \log p(\boldsymbol{z}, \boldsymbol{y}|\boldsymbol{x})\mathrm{d}\boldsymbol{z}\mathrm{d}\boldsymbol{y} \\ &= \int\int p(\boldsymbol{z}, \boldsymbol{y}|\boldsymbol{x}) \log \frac{p(\boldsymbol{y}|\boldsymbol{x})p(\boldsymbol{z}, \boldsymbol{y}|\boldsymbol{x})}{p(\boldsymbol{y}|\boldsymbol{x})p(\boldsymbol{z}|\boldsymbol{x})}\mathrm{d}\boldsymbol{y}\mathrm{d}\boldsymbol{z} \\ &= D_{\mathrm{KL}}\left(p(\boldsymbol{y}, \boldsymbol{z}|\boldsymbol{x})\|p(\boldsymbol{y}|\boldsymbol{x})p(\boldsymbol{z}|\boldsymbol{x})\right) + \int\int p(\boldsymbol{z}, \boldsymbol{y}|\boldsymbol{x}) \log p(\boldsymbol{y}|\boldsymbol{x})\mathrm{d}\boldsymbol{z}\mathrm{d}\boldsymbol{y} \\ &= D_{\mathrm{KL}}\left(p(\boldsymbol{y}, \boldsymbol{z}|\boldsymbol{x})\|p(\boldsymbol{y}|\boldsymbol{x})p(\boldsymbol{z}|\boldsymbol{x})\right) + \int p(\boldsymbol{y}|\boldsymbol{x}) \log p(\boldsymbol{y}|\boldsymbol{x})\mathrm{d}\boldsymbol{y} \\ &= D_{\mathrm{KL}}\left(p(\boldsymbol{y}, \boldsymbol{z}|\boldsymbol{x})\|p(\boldsymbol{y}|\boldsymbol{x})p(\boldsymbol{z}|\boldsymbol{x})\right) + \int\int p(\boldsymbol{y}|\boldsymbol{x})p(\boldsymbol{z}|\boldsymbol{x}) \log p(\boldsymbol{z}|\boldsymbol{x})\mathrm{d}\boldsymbol{z}\mathrm{d}\boldsymbol{y} \\ &= D_{\mathrm{KL}}\left(p(\boldsymbol{y}, \boldsymbol{z}|\boldsymbol{x})\|p(\boldsymbol{y}|\boldsymbol{x})p(\boldsymbol{z}|\boldsymbol{x})\right) + \int\int p(\boldsymbol{y}|\boldsymbol{x})p(\boldsymbol{z}|\boldsymbol{x}) \log \frac{p(\boldsymbol{z}|\boldsymbol{x})p(\boldsymbol{y}|\boldsymbol{x})p(\boldsymbol{z}, \boldsymbol{y}|\boldsymbol{x})}{p(\boldsymbol{z}, \boldsymbol{y}|\boldsymbol{x})p(\boldsymbol{z}|\boldsymbol{x})}\mathrm{d}\boldsymbol{z}\mathrm{d}\boldsymbol{y} \\ &= D_{\mathrm{KL}}\left(p(\boldsymbol{y}, \boldsymbol{z}|\boldsymbol{x})\|p(\boldsymbol{y}|\boldsymbol{x})p(\boldsymbol{z}|\boldsymbol{x})\right) + D_{\mathrm{KL}}\left(p(\boldsymbol{y}|\boldsymbol{x})p(\boldsymbol{z}|\boldsymbol{x})\|p(\boldsymbol{y}, \boldsymbol{z}|\boldsymbol{x})\right) \\ &\quad + \mathbb{E}_{p(\boldsymbol{z}|\boldsymbol{x})p(\boldsymbol{y}|\boldsymbol{x})}[\log p(\boldsymbol{y}|\boldsymbol{z}, \boldsymbol{x})] \\ &\geq \mathbb{E}_{p(\boldsymbol{z}|\boldsymbol{x})p(\boldsymbol{y}|\boldsymbol{x})}[\log p(\boldsymbol{y}|\boldsymbol{z}, \boldsymbol{x})]. \end{aligned} \tag{20}$$

Plugging Eq. (20) into Eq. (19) and using $p(\boldsymbol{y}|\boldsymbol{z}) = p(\boldsymbol{y}|\boldsymbol{z}, \boldsymbol{x})$ again, we obtain:

$$
\begin{aligned}
I(Z, Y) &= \mathbb{E}_{p(\boldsymbol{x})} \mathbb{E}_{p(\boldsymbol{y}|\boldsymbol{x})} \mathbb{E}_{p(\boldsymbol{z}|\boldsymbol{x}, \boldsymbol{y})} [\log p(\boldsymbol{y}|\boldsymbol{z}, \boldsymbol{x})] + H(Y) \\
&= \mathbb{E}_{p(\boldsymbol{x})} \mathbb{E}_{p(\boldsymbol{y}|\boldsymbol{z})} \mathbb{E}_{p(\boldsymbol{z}|\boldsymbol{x})} [\log p(\boldsymbol{y}|\boldsymbol{z})] \\
&\quad + \mathbb{E}_{p(\boldsymbol{x})} \left[ D_{\mathrm{KL}} \left( p(\boldsymbol{y}, \boldsymbol{z}|\boldsymbol{x}) \| p(\boldsymbol{y}|\boldsymbol{x}) p(\boldsymbol{z}|\boldsymbol{x}) \right) \right] \\
&\quad + \mathbb{E}_{p(\boldsymbol{x})} \left[ D_{\mathrm{KL}} \left( p(\boldsymbol{y}|\boldsymbol{x}) p(\boldsymbol{z}|\boldsymbol{x}) \| p(\boldsymbol{y}, \boldsymbol{z}|\boldsymbol{x}) \right) \right] + H(Y) \\
&= \mathbb{E}_{p(\boldsymbol{x})} \mathbb{E}_{p(\boldsymbol{y}|\boldsymbol{z})} \mathbb{E}_{p(\boldsymbol{z}|\boldsymbol{x})} [\log p(\boldsymbol{y}|\boldsymbol{z})] \\
&\quad + I(Y, Z|X) + L(Y, Z|X) + H(Y) \\
&\geq \mathbb{E}_{p(\boldsymbol{x})} \mathbb{E}_{p(\boldsymbol{y}|\boldsymbol{x})} \mathbb{E}_{p(\boldsymbol{z}|\boldsymbol{x})} [\log p(\boldsymbol{y}|\boldsymbol{z})] + H(Y) \\
&\geq \mathbb{E}_{p(\boldsymbol{x})} \mathbb{E}_{p(\boldsymbol{y}|\boldsymbol{x})} \mathbb{E}_{p(\boldsymbol{z}|\boldsymbol{x})} [\log p(\boldsymbol{y}|\boldsymbol{z})],
\end{aligned}
\tag{21}
$$

where $I(Y, Z|X)$ is the conditional mutual information, and $L(Y, Z|X)$ is the conditional lautum information (Palomar & Verdu, 2008), respectively.

## B    PROOF OF PROPOSITION 1

**A brief introduction to VB-based EDL methods:** Some previous EDL methods (Chen et al., 2018; Joo et al., 2020) derive the optimization objective from the variational Bayes (VB) perspective, where the goal is to optimize the model's parameters $\boldsymbol{\theta}$ such that the *posterior* distribution $p(\boldsymbol{\pi}|\boldsymbol{x};\boldsymbol{\theta})$ aligns with the *true* posterior distribution $p(\boldsymbol{\pi}|\boldsymbol{y})$.

For notation simplicity, we omit the model's parameters $\boldsymbol{\theta}$ in $p(\boldsymbol{\pi}|\boldsymbol{x};\boldsymbol{\theta})$ (or $p(\boldsymbol{z}|\boldsymbol{x};\boldsymbol{\theta})$) and use $p(\boldsymbol{\pi}|\boldsymbol{x})$ (or $p(\boldsymbol{z}|\boldsymbol{x})$) instead.

**Remark 1.** *The condition in Proposition 1 is that the latent variable $\boldsymbol{z} = \boldsymbol{\pi}$, and the prior $r(\boldsymbol{z}) = r(\boldsymbol{\pi}) = Dir\left(\boldsymbol{\pi}; \boldsymbol{y} \odot \boldsymbol{\alpha} + (\mathbf{1} - \boldsymbol{y})\right)$. In fact, the choice of the prior $r(\boldsymbol{\pi})$ is not unique. For example, Chen et al. (2018) present three options. The correctness of Proposition 1 remains unaffected by the choice of prior. Proposition 1 uses one exemplary prior $r(\boldsymbol{\pi}) = Dir\left(\boldsymbol{\pi}; \boldsymbol{y} \odot \boldsymbol{\alpha} + (\mathbf{1} - \boldsymbol{y})\right)$ suggested by Chen et al. (2018).*

*Proof.* The target of VB-based EDL methods is:

$$\min_{\boldsymbol{\theta}} \mathbb{E}_{p(\boldsymbol{x})} \mathbb{E}_{p(\boldsymbol{y}|\boldsymbol{x})} \left[D_{\text{KL}}\left(p(\boldsymbol{\pi}|\boldsymbol{x})\|p(\boldsymbol{\pi}|\boldsymbol{y})\right)\right]. \tag{22}$$

Next, we show that the IB objective in Eq. (10) is an upper bound of Eq. (22) when $\boldsymbol{z} = \boldsymbol{\pi}$ and $r(\boldsymbol{z}) = r(\boldsymbol{\pi})$ for a given prior $r(\boldsymbol{\pi})$, e.g., $r(\boldsymbol{\pi}) = \text{Dir}\left(\boldsymbol{\pi}; \boldsymbol{y} \odot \boldsymbol{\alpha} + (\mathbf{1} - \boldsymbol{y})\right)$. Minimizing the IB objective in Eq. (10) therefore provides a tractable way to approximate the minimization of Eq. (22).

If we choose $\beta = 1$, $\boldsymbol{z} = \boldsymbol{\pi}$ and a prior $r(\boldsymbol{z}) = r(\boldsymbol{\pi})$, then the IB objective in Eq. (10) becomes

$$\min_{\boldsymbol{\theta}} -\mathbb{E}_{p(\boldsymbol{x})} \mathbb{E}_{p(\boldsymbol{y}|\boldsymbol{x})} \mathbb{E}_{p(\boldsymbol{\pi}|\boldsymbol{x})} \left[\log p(\boldsymbol{y}|\boldsymbol{\pi})\right] + \mathbb{E}_{p(\boldsymbol{x})} \left[D_{\text{KL}}\left(p(\boldsymbol{\pi}|\boldsymbol{x})\|r(\boldsymbol{\pi})\right)\right]. \tag{23}$$

Expanding Eq. (23), we have:

$$\mathbb{E}_{p(\boldsymbol{x})} \left[D_{\text{KL}}\left(p(\boldsymbol{\pi}|\boldsymbol{x})\|r(\boldsymbol{\pi})\right)\right] - \mathbb{E}_{p(\boldsymbol{x})} \mathbb{E}_{p(\boldsymbol{y}|\boldsymbol{x})} \mathbb{E}_{p(\boldsymbol{\pi}|\boldsymbol{x})} \left[\log p(\boldsymbol{y}|\boldsymbol{\pi})\right]$$

$$= \mathbb{E}_{p(\boldsymbol{x},\boldsymbol{y})} \left[D_{\text{KL}}\left(p(\boldsymbol{\pi}|\boldsymbol{x})\|r(\boldsymbol{\pi})\right)\right] - \mathbb{E}_{p(\boldsymbol{x})} \mathbb{E}_{p(\boldsymbol{y}|\boldsymbol{x})} \mathbb{E}_{p(\boldsymbol{\pi}|\boldsymbol{x})} \left[\log p(\boldsymbol{y}|\boldsymbol{\pi})\right]$$

$$= \mathbb{E}_{p(\boldsymbol{x})} \mathbb{E}_{p(\boldsymbol{y}|\boldsymbol{x})} \left[\int p(\boldsymbol{\pi}|\boldsymbol{x}) \log \frac{p(\boldsymbol{\pi}|\boldsymbol{x})}{r(\boldsymbol{\pi})} d\boldsymbol{\pi}\right] - \mathbb{E}_{p(\boldsymbol{x})} \mathbb{E}_{p(\boldsymbol{y}|\boldsymbol{x})} \left[\int p(\boldsymbol{\pi}|\boldsymbol{x}) \log p(\boldsymbol{y}|\boldsymbol{\pi}) d\boldsymbol{\pi}\right]$$

$$\geq \underbrace{\mathbb{E}_{p(\boldsymbol{x})} \mathbb{E}_{p(\boldsymbol{y}|\boldsymbol{x})} \left[\int p(\boldsymbol{\pi}|\boldsymbol{x}) \log \frac{p(\boldsymbol{\pi}|\boldsymbol{x})}{p(\boldsymbol{\pi})} d\boldsymbol{\pi}\right]}_{\text{Use } D_{\text{KL}}(p(\boldsymbol{\pi})\|r(\boldsymbol{\pi})) \geq 0 \text{ (Similar to Eq. (18))}} - \mathbb{E}_{p(\boldsymbol{x})} \mathbb{E}_{p(\boldsymbol{y}|\boldsymbol{x})} \left[\int p(\boldsymbol{\pi}|\boldsymbol{x}) \log p(\boldsymbol{y}|\boldsymbol{\pi}) d\boldsymbol{\pi}\right]$$

$$\geq \mathbb{E}_{p(\boldsymbol{x})} \mathbb{E}_{p(\boldsymbol{y}|\boldsymbol{x})} \left[\int p(\boldsymbol{\pi}|\boldsymbol{x}) \log \frac{p(\boldsymbol{\pi}|\boldsymbol{x})}{p(\boldsymbol{\pi})} d\boldsymbol{\pi}\right] - \mathbb{E}_{p(\boldsymbol{x})} \mathbb{E}_{p(\boldsymbol{y}|\boldsymbol{x})} \left[\int p(\boldsymbol{\pi}|\boldsymbol{x}) \log p(\boldsymbol{y}|\boldsymbol{\pi}) d\boldsymbol{\pi}\right] \underbrace{-H(Y)}_{\leq 0}$$

$$= \mathbb{E}_{p(\boldsymbol{x})} \mathbb{E}_{p(\boldsymbol{y}|\boldsymbol{x})} \left[\int p(\boldsymbol{\pi}|\boldsymbol{x}) \log \frac{p(\boldsymbol{\pi}|\boldsymbol{x})}{p(\boldsymbol{y}|\boldsymbol{\pi})p(\boldsymbol{\pi})} d\boldsymbol{\pi}\right] + \mathbb{E}_{p(\boldsymbol{y})} \left[\log p(\boldsymbol{y})\right]$$

$$= \mathbb{E}_{p(\boldsymbol{x})} \mathbb{E}_{p(\boldsymbol{y}|\boldsymbol{x})} \left[\int p(\boldsymbol{\pi}|\boldsymbol{x}) \log \frac{p(\boldsymbol{\pi}|\boldsymbol{x})}{p(\boldsymbol{y}|\boldsymbol{\pi})p(\boldsymbol{\pi})} d\boldsymbol{\pi}\right] + \mathbb{E}_{p(\boldsymbol{x})} \mathbb{E}_{p(\boldsymbol{y}|\boldsymbol{x})} \left[\log p(\boldsymbol{y})\right]$$

$$= \mathbb{E}_{p(\boldsymbol{x})} \mathbb{E}_{p(\boldsymbol{y}|\boldsymbol{x})} \left[\int p(\boldsymbol{\pi}|\boldsymbol{x}) \log \frac{p(\boldsymbol{\pi}|\boldsymbol{x})p(\boldsymbol{y})}{p(\boldsymbol{\pi},\boldsymbol{y})} d\boldsymbol{\pi}\right]$$

$$= \mathbb{E}_{p(\boldsymbol{x})} \mathbb{E}_{p(\boldsymbol{y}|\boldsymbol{x})} \left[\int p(\boldsymbol{\pi}|\boldsymbol{x}) \log \frac{p(\boldsymbol{\pi}|\boldsymbol{x})}{p(\boldsymbol{\pi}|\boldsymbol{y})} d\boldsymbol{\pi}\right]$$

$$= \mathbb{E}_{p(\boldsymbol{x})} \mathbb{E}_{p(\boldsymbol{y}|\boldsymbol{x})} \left[D_{\text{KL}}\left(p(\boldsymbol{\pi}|\boldsymbol{x})\|p(\boldsymbol{\pi}|\boldsymbol{y})\right)\right],$$

$$\tag{24}$$

which is the target of VB-based EDL methods.   □

Table 5: Loss weight $\beta$ for IB-EDL.

| Model | ID Calibration | | | | | |
| --- | --- | --- | --- | --- | --- | --- |
| | ARC-C | ARC-E | OBQA | CSQA | SciQ | RACE |
| Llama2-7B | $1 \times 10^{-3}$ | $1 \times 10^{-4}$ | $1.5 \times 10^{-4}$ | $5 \times 10^{-5}$ | $1 \times 10^{-6}$ | $1 \times 10^{-6}$ |
| Llama3-8B | $9 \times 10^{-5}$ | $2 \times 10^{-6}$ | $1 \times 10^{-5}$ | $3 \times 10^{-5}$ | $5 \times 10^{-7}$ | $1 \times 10^{-6}$ |
| Mistral-7B | $9 \times 10^{-5}$ | $1 \times 10^{-6}$ | $1 \times 10^{-5}$ | $5 \times 10^{-5}$ | $4 \times 10^{-6}$ | $2 \times 10^{-6}$ |
| Model | Learning with Label Noise | | | | | |
| | ARC-C | ARC-E | OBQA | CSQA | SciQ | RACE |
| Llama2-7B | $2 \times 10^{-3}$ | $1 \times 10^{-4}$ | $5 \times 10^{-4}$ | $1.6 \times 10^{-4}$ | $5 \times 10^{-6}$ | $1 \times 10^{-6}$ |
| Llama3-8B | $5 \times 10^{-6}$ | $2 \times 10^{-5}$ | $6 \times 10^{-5}$ | $3 \times 10^{-5}$ | $5 \times 10^{-6}$ | $2 \times 10^{-6}$ |
| Mistral-7B | $1 \times 10^{-5}$ | $2 \times 10^{-5}$ | $9 \times 10^{-5}$ | $1 \times 10^{-5}$ | $1 \times 10^{-6}$ | $7 \times 10^{-6}$ |

## C  DETAILED IMPLEMENTATION

**Models:** We fine-tuned three models: Llama2-7B (Touvron et al., 2023), Llama3-8B (Dubey et al., 2024), and Mistral-7B-v0.1 (Jiang et al., 2023).

**LoRA hyperparameters:** We applied LoRA (Hu et al., 2022) finetuning using the PEFT (Mangrulkar et al., 2022) library. For all models, LoRA adaptors were applied to the `q_proj`, `v_proj`, and `lm_head` modules. Additionally, we used Dropout with a dropout rate of $p = 0.1$, LoRA $\alpha = 16$, rank $r = 8$, and set `bias = "lora_only"`.

**Training details:** For the MAP baseline and all EDL methods, all models were trained for 30000 steps on the CSQA dataset and 10080 steps on the other datasets. The learning rate was set to 0.00005 and annealed using a cosine schedule. The maximum token length was set to 300 for the RACE dataset and 256 for all other datasets. Training was conducted with `bfloat16` precision. For MCD (Gal & Ghahramani, 2016), we performed 10 forward passes. For Ens (Lakshminarayanan et al., 2017; Fort et al., 2019), we used predictions from 3 models. For LA (Yang et al., 2024), I-EDL (Deng et al., 2023), and R-EDL (Chen et al., 2024), we used the original implementations and hyperparameters; please refer to the respective official codebases. For EDL methods, we follow the previous works to apply gradient clipping with maximal gradient norm of 20 to stabilize the training.

**IB-EDL implementation details:** Both linear prediction heads (for $\boldsymbol{\mu}$ and $\boldsymbol{\sigma}$) are initialized using the linear head from the pretrained LLM. Additionally, both heads are equipped with LoRA adapters and are jointly trained alongside the LoRA weights of the transformer encoder layers. By default, we used $K = 20$ for sampling pre-evidences from the predicted Gaussian distribution during both training and inference. Table 5 lists the $\beta$ values used in different experiments. Note that for OOD detection experiments, the parameters are identical to those used for OBQA, as OBQA is the training set for these experiments. We optimized $\beta$ values using grid search. We suggest a guideline for selecting $\beta$: if the MAP baseline shows lower accuracy and higher ECE, or in the presence of label noise, a larger $\beta$ may be chosen to encourage stronger compression and forgetting of the input. Otherwise, a smaller $\beta$ can be selected to allow more information from the input follow through the pre-evidences.

## D  A POST-HOC CALIBRATION TECHNIQUE FOR IB-EDL

In this section, we introduce an additional *post-hoc* calibration technique aimed at further refining the calibration of the IB-EDL finetuned model.

**Intuition:** The key motivation behind this calibration method is to account for an additional source of uncertainty in IB-EDL: the variance $\sigma_j^2$ of the pre-evidences. In IB-EDL, the uncertainty mass and belief mass are normalized by the constraint: $\sum_j b_j + u = 1$, where $b_j = \frac{\alpha_j - 1}{\sum_j \alpha_j}$. To incorporate the additional uncertainty arising from the variance of pre-evidences, we consider the effect of increasing $u$, which leads to a reduction in $\sum_j b_j$. This suggests the need to adjust $\alpha_j$ by

Table 6: Calibration performance of uncertainty-aware methods on fine-tuned Mistral-7B.

| Metrics | Method | ARC-C | ARC-E | OBQA | CSQA | SciQ | RACE |
|---|---|---|---|---|---|---|---|
| Acc ↑ | MAP | $80.19_{\pm0.68}$ | $92.07_{\pm0.12}$ | $88.06_{\pm0.61}$ | $83.24_{\pm1.47}$ | $\underline{94.73}_{\pm0.06}$ | $86.61_{\pm0.28}$ |
| | MCD | $80.26_{\pm0.52}$ | $92.13_{\pm0.15}$ | $88.07_{\pm0.61}$ | $83.24_{\pm1.47}$ | $\underline{94.73}_{\pm0.06}$ | $\underline{86.64}_{\pm0.22}$ |
| | Ens | $80.48_{\pm0.45}$ | $\underline{92.41}_{\pm0.12}$ | $\underline{88.51}_{\pm0.18}$ | $\underline{83.41}_{\pm1.44}$ | $\mathbf{94.80}_{\pm0.10}$ | $\mathbf{86.81}_{\pm0.21}$ |
| | LA | $78.47_{\pm1.10}$ | $92.12_{\pm0.11}$ | $88.19_{\pm0.69}$ | $83.21_{\pm1.01}$ | $94.66_{\pm0.16}$ | $86.34_{\pm0.39}$ |
| | EDL | $\mathbf{80.60}_{\pm0.47}$ | $92.27_{\pm0.44}$ | $87.23_{\pm1.42}$ | $83.31_{\pm0.34}$ | $94.27_{\pm0.23}$ | $85.70_{\pm0.41}$ |
| | VID | $80.37_{\pm0.39}$ | $91.85_{\pm0.13}$ | $88.33_{\pm0.81}$ | $82.52_{\pm0.54}$ | $94.17_{\pm0.12}$ | $86.40_{\pm0.21}$ |
| | I-EDL | $80.46_{\pm1.63}$ | $92.28_{\pm0.15}$ | $88.06_{\pm0.31}$ | $82.91_{\pm0.20}$ | $94.03_{\pm0.32}$ | $85.11_{\pm0.61}$ |
| | R-EDL | $80.12_{\pm0.97}$ | $92.20_{\pm0.15}$ | $88.33_{\pm0.91}$ | $83.07_{\pm0.83}$ | $94.03_{\pm0.31}$ | $86.07_{\pm0.28}$ |
| | IB-EDL | $80.57_{\pm0.68}$ | $\mathbf{92.47}_{\pm0.12}$ | $\mathbf{88.73}_{\pm0.70}$ | $\mathbf{83.65}_{\pm0.45}$ | $94.40_{\pm0.26}$ | $86.40_{\pm0.32}$ |
| ECE ↓ | MAP | $19.42_{\pm0.68}$ | $7.63_{\pm0.16}$ | $11.29_{\pm0.31}$ | $15.72_{\pm1.46}$ | $4.98_{\pm0.06}$ | $8.43_{\pm0.46}$ |
| | MCD | $19.26_{\pm0.52}$ | $7.62_{\pm0.16}$ | $11.22_{\pm0.40}$ | $15.72_{\pm1.46}$ | $4.98_{\pm0.06}$ | $8.42_{\pm0.47}$ |
| | Ens | $17.04_{\pm1.58}$ | $7.03_{\pm0.79}$ | $8.87_{\pm0.90}$ | $14.18_{\pm2.52}$ | $4.78_{\pm0.13}$ | $8.21_{\pm0.62}$ |
| | LA | $20.04_{\pm0.62}$ | $\mathbf{1.57}_{\pm0.39}$ | $\underline{4.49}_{\pm0.17}$ | $15.11_{\pm2.95}$ | $\mathbf{1.91}_{\pm0.46}$ | $\mathbf{2.94}_{\pm0.25}$ |
| | EDL | $6.21_{\pm1.18}$ | $6.25_{\pm0.46}$ | $6.23_{\pm0.56}$ | $7.71_{\pm0.86}$ | $7.64_{\pm0.85}$ | $7.75_{\pm0.86}$ |
| | VID | $9.38_{\pm0.46}$ | $\underline{2.44}_{\pm0.16}$ | $4.88_{\pm0.77}$ | $7.32_{\pm0.74}$ | $5.62_{\pm0.52}$ | $4.20_{\pm0.13}$ |
| | I-EDL | $\underline{4.70}_{\pm2.15}$ | $10.94_{\pm1.27}$ | $9.63_{\pm0.75}$ | $8.85_{\pm0.44}$ | $11.00_{\pm0.30}$ | $13.16_{\pm0.87}$ |
| | R-EDL | $11.26_{\pm0.18}$ | $2.86_{\pm1.07}$ | $5.43_{\pm0.65}$ | $\underline{6.30}_{\pm0.65}$ | $\underline{4.48}_{\pm0.23}$ | $3.69_{\pm0.52}$ |
| | IB-EDL | $\mathbf{3.60}_{\pm1.10}$ | $3.49_{\pm1.24}$ | $\mathbf{2.27}_{\pm0.67}$ | $\mathbf{6.02}_{\pm0.08}$ | $4.99_{\pm1.22}$ | $\underline{3.58}_{\pm0.22}$ |
| NLL ↓ | MAP | $2.18_{\pm0.14}$ | $0.85_{\pm0.03}$ | $0.85_{\pm0.02}$ | $1.18_{\pm0.06}$ | $0.31_{\pm0.01}$ | $0.52_{\pm0.01}$ |
| | MCD | $2.18_{\pm0.13}$ | $0.84_{\pm0.04}$ | $0.84_{\pm0.03}$ | $1.18_{\pm0.07}$ | $0.31_{\pm0.01}$ | $0.52_{\pm0.02}$ |
| | Ens | $1.82_{\pm0.17}$ | $0.78_{\pm0.10}$ | $0.67_{\pm0.04}$ | $1.01_{\pm0.13}$ | $0.28_{\pm0.01}$ | $0.51_{\pm0.03}$ |
| | LA | $0.78_{\pm0.02}$ | $\mathbf{0.29}_{\pm0.01}$ | $\mathbf{0.38}_{\pm0.02}$ | $0.59_{\pm0.03}$ | $\mathbf{0.17}_{\pm0.01}$ | $0.45_{\pm0.03}$ |
| | EDL | $\underline{0.71}_{\pm0.04}$ | $0.35_{\pm0.02}$ | $0.47_{\pm0.03}$ | $0.63_{\pm0.01}$ | $0.27_{\pm0.01}$ | $0.47_{\pm0.02}$ |
| | VID | $0.76_{\pm0.03}$ | $0.36_{\pm0.01}$ | $0.46_{\pm0.01}$ | $0.68_{\pm0.02}$ | $0.24_{\pm0.01}$ | $\underline{0.41}_{\pm0.01}$ |
| | I-EDL | $\underline{0.71}_{\pm0.05}$ | $0.38_{\pm0.02}$ | $0.45_{\pm0.01}$ | $0.65_{\pm0.01}$ | $0.29_{\pm0.01}$ | $0.50_{\pm0.01}$ |
| | R-EDL | $0.77_{\pm0.02}$ | $0.33_{\pm0.01}$ | $\underline{0.41}_{\pm0.03}$ | $0.64_{\pm0.01}$ | $0.24_{\pm0.01}$ | $0.45_{\pm0.01}$ |
| | IB-EDL | $\mathbf{0.70}_{\pm0.01}$ | $\underline{0.32}_{\pm0.02}$ | $\underline{0.41}_{\pm0.01}$ | $\underline{0.61}_{\pm0.01}$ | $\underline{0.23}_{\pm0.01}$ | $\mathbf{0.40}_{\pm0.01}$ |

subtracting a term that is proportional to the standard deviation $\sigma_j$, which is predicted by IB-EDL. We parameterize this adjustment as $\zeta \cdot \sigma_j$, where $\zeta$ is a scalar hyperparameter. Consequently, we update $\alpha_j$ as follows: $\alpha_j \leftarrow \alpha_j - \zeta \cdot \sigma_j$. The intuition behind this update is that if the model exhibits high uncertainty in predicting $\alpha_j$ (i.e., predicting pre-evidences $\tilde{e}_j$), we enforce a more conservative belief representation by reducing $\alpha_j$ accordingly. This adjustment prevents excessively large values of $\alpha_j$ and ensures a more calibrated uncertainty estimation. As a result, the updated uncertainty mass is given by: $u = \frac{C}{\sum_j \alpha_j - \zeta \cdot \sum_j \sigma_j}$. This approach effectively integrates the variance of pre-evidences into the calibration process, leading to improved uncertainty quantification in IB-EDL.

**Determining the optimal value of** $\zeta$**:** Theoretically, the EDL model can still be overconfident or underconfident after training, leading to an overestimation or underestimation of the uncertainty mass $u$. Since $u$ may require adjustment in either direction, we allow the hyperparameter $\zeta \in \mathbb{R}$ to be either positive or negative, enabling calibration in both cases. To determine the optimal $\zeta$, we recommend to analyze the calibration curve (Guo et al., 2017) on the training or validation set, which plots accuracy against confidence for binned samples. If the calibration curve lies below the optimal diagonal line, it indicates that the model is overconfident. In this case, we set $\zeta > 0$ to encourage greater uncertainty and improve calibration. Conversely, if the calibration curve lies above the diagonal, the model is underconfident, and we set $\zeta < 0$ to increase certainty and correct for underconfidence.

**Values of** $\zeta$ **used in additional experiments:** It is important to note that this post-hoc calibration technique is an optional component of IB-EDL and is not required in all cases. In the experiments presented in the main text, we omit this technique. In the calibration experiment on OOD test sets presented in Table 12, we use $\zeta = -1.0$ for OBQA $\rightarrow$ ARC-C, $\zeta = 3.0$ for OBQA $\rightarrow$ ARC-E, and $\zeta = -5.0$ for OBQA $\rightarrow$ CSQA.

Table 7: OOD detection performance on fine-tuned Mistral-7B. $A \rightarrow B$ indicates $A$ as the ID training set and $B$ as the OOD test set. MP and UM are two scores for measuring AUROC.

| Model | Method | OBQA → ARC-C | | OBQA → ARC-E | | OBQA → CSQA | |
| --- | --- | --- | --- | --- | --- | --- | --- |
| | | AUROC ↑ | | AUROC ↑ | | AUROC ↑ | |
| | | MP | UM | MP | UM | MP | UM |
| Mistral-7B | MAP | $60.40_{\pm 1.36}$ | – | $53.30_{\pm 1.16}$ | – | $63.70_{\pm 1.10}$ | – |
| | MCD | $60.39_{\pm 1.37}$ | – | $53.30_{\pm 1.16}$ | – | $63.70_{\pm 1.10}$ | – |
| | Ens | $60.67_{\pm 0.87}$ | – | $54.05_{\pm 0.88}$ | – | $63.80_{\pm 1.09}$ | – |
| | LA | $61.61_{\pm 1.05}$ | – | $53.14_{\pm 0.93}$ | – | $68.31_{\pm 1.11}$ | – |
| | EDL | $84.17_{\pm 1.87}$ | $77.34_{\pm 6.97}$ | $81.81_{\pm 2.31}$ | $74.18_{\pm 6.38}$ | $83.55_{\pm 2.67}$ | $78.28_{\pm 5.81}$ |
| | VID | $86.30_{\pm 3.85}$ | $\underline{84.74}_{\pm 0.89}$ | $\underline{88.16}_{\pm 1.87}$ | $\underline{90.68}_{\pm 0.17}$ | $\underline{88.93}_{\pm 1.39}$ | $81.45_{\pm 2.04}$ |
| | I-EDL | $85.02_{\pm 0.90}$ | $82.28_{\pm 1.44}$ | $82.67_{\pm 1.34}$ | $79.42_{\pm 2.33}$ | $85.08_{\pm 0.87}$ | $\underline{82.07}_{\pm 1.43}$ |
| | R-EDL | $76.26_{\pm 1.05}$ | $72.85_{\pm 0.61}$ | $71.95_{\pm 0.78}$ | $67.56_{\pm 0.72}$ | $75.59_{\pm 1.32}$ | $71.93_{\pm 2.05}$ |
| | IB-EDL | $\mathbf{90.28}_{\pm 1.22}$ | $\mathbf{88.53}_{\pm 1.08}$ | $\mathbf{89.54}_{\pm 1.16}$ | $\mathbf{94.29}_{\pm 0.49}$ | $\mathbf{90.45}_{\pm 1.13}$ | $\mathbf{83.85}_{\pm 2.55}$ |

# E  ADDITIONAL EXPERIMENTAL RESULTS

## E.1  ID CALIBRATION AND OOD DETECTION RESULTS OF MISTRAL-7B

In this subsection, we present the ID calibration and OOD detection performance of the fine-tuned Mistral-7B model. Table 6 provides an overview of the calibration performance on ID datasets, while Table 7 reports the OOD detection results.

## E.2  ADDITIONAL RESULTS OF FINE-TUNING MODELS ON NOISY DATASETS

Table 8 and Table 9, and Table 10 show the results of fine-tuning Llama2-7B, Llama3-8B, and Mistral-7B on noisy datasets, respectively.

**Results:** In the context of noisy data, the primary objective is to preserve accuracy. Therefore, in Section 4.4 of the main text, we focus primarily on accuracy results. As highlighted in Section 4.2, a good uncertainty-aware method should achieve comparable accuracy while maintaining a low calibration error, a position shared by many other current research efforts (see, e.g., Chen et al., 2024). Here, we thus provide additional results for the ECE and NLL metrics. In the presence of label noise, Non-EDL methods tend to underperform EDL methods significantly in terms of accuracy. In the comparisons of Tables 8 to 10, we therefore focus on the comparison of EDL methods, which manage to strike the right balance of good accuracy and uncertainty quantification. Among these, IB-EDL demonstrates the highest accuracy and achieves the lowest ECE and NLL compared to other EDL approaches, highlighting its robustness against label noise.

## E.3  ADDITIONAL OOD DETECTION RESULTS ON MMLU-MATH DATASET

We conduct additional OOD detection experiments using Llama2-7B in a setting characterized by a significant distribution shift. For this purpose, we select a subset of the MMLU dataset (Hendrycks et al., 2021a;b) as the OOD test set, which includes data samples from the math-related topics `college_mathematics`, `high_school_mathematics`, and `abstract_algebra`. We refer to this subset as the MMLU-Math dataset. In addition, we use the OBQA dataset as the ID dataset. While OBQA focuses on common-sense reasoning, the MMLU-Math dataset requires advanced mathematical knowledge to solve its questions. As a result, the distribution shift in this setting is substantially larger compared to settings like OBQA → ARC-C.

As shown in Table 11, IB-EDL consistently achieves the best OOD detection performance when using either MP or UM as the detection score. Additionally, a general trend is observed: the AUROCs of IB-EDL and other baselines improve as the distribution shift increases compared to the setting OBQA → ARC-C.

Table 8: Fine-tuning Llama2-7B on noisy datasets. In each training dataset, the labels of 30% samples are randomly perturbed. A robust uncertainty-aware method should not only maintain accuracy but also exhibit low calibration error. Non-EDL methods tend to significantly underperform EDL methods in terms of Accuracy. Therefore, the comparison for ECE and NLL is limited to EDL methods, with the best and second-best values among them highlighted.

| Metrics | Method | ARC-C | ARC-E | OBQA | CSQA | SciQ | RACE |
|---|---|---|---|---|---|---|---|
| Acc ↑ | MAP | $51.08_{\pm2.96}$ | $71.39_{\pm2.73}$ | $73.93_{\pm2.21}$ | $74.56_{\pm0.31}$ | $88.63_{\pm0.06}$ | $76.83_{\pm0.39}$ |
| | MCD | $51.08_{\pm2.96}$ | $71.40_{\pm2.72}$ | $73.93_{\pm2.20}$ | $74.56_{\pm0.32}$ | $88.64_{\pm0.06}$ | $76.95_{\pm0.19}$ |
| | Ens | $56.10_{\pm3.11}$ | $76.03_{\pm1.99}$ | $75.23_{\pm1.06}$ | $74.59_{\pm0.23}$ | $88.64_{\pm0.06}$ | $76.94_{\pm0.22}$ |
| | LA | $53.38_{\pm2.17}$ | $74.90_{\pm1.76}$ | $74.57_{\pm2.22}$ | $74.59_{\pm0.20}$ | $88.47_{\pm0.21}$ | $77.04_{\pm0.39}$ |
| | EDL | $57.16_{\pm3.33}$ | $76.16_{\pm1.12}$ | $74.46_{\pm1.33}$ | $74.65_{\pm0.05}$ | $89.03_{\pm0.25}$ | $77.69_{\pm0.44}$ |
| | VID | $\underline{57.56}_{\pm0.58}$ | $76.57_{\pm1.79}$ | $76.67_{\pm1.42}$ | $\underline{75.97}_{\pm0.66}$ | $89.06_{\pm0.21}$ | $\underline{78.75}_{\pm0.25}$ |
| | I-EDL | $53.86_{\pm1.25}$ | $76.08_{\pm0.71}$ | $\underline{77.00}_{\pm0.88}$ | $74.01_{\pm1.28}$ | $\mathbf{89.26}_{\pm\mathbf{0.32}}$ | $76.48_{\pm0.87}$ |
| | R-EDL | $57.00_{\pm0.76}$ | $\underline{76.96}_{\pm1.07}$ | $73.20_{\pm1.44}$ | $74.36_{\pm0.49}$ | $87.33_{\pm0.49}$ | $78.01_{\pm0.53}$ |
| | IB-EDL | $\mathbf{59.06}_{\pm\mathbf{2.07}}$ | $\mathbf{80.27}_{\pm\mathbf{0.48}}$ | $\mathbf{78.13}_{\pm\mathbf{0.99}}$ | $\mathbf{76.35}_{\pm\mathbf{0.66}}$ | $\underline{89.06}_{\pm0.50}$ | $\mathbf{79.41}_{\pm\mathbf{0.59}}$ |
| ECE ↓ | MAP | $21.89_{\pm0.11}$ | $8.95_{\pm1.04}$ | $8.21_{\pm0.78}$ | $4.45_{\pm1.51}$ | $19.05_{\pm0.17}$ | $15.87_{\pm0.30}$ |
| | MCD | $21.87_{\pm0.12}$ | $8.91_{\pm1.10}$ | $8.23_{\pm0.82}$ | $4.47_{\pm1.55}$ | $19.04_{\pm0.16}$ | $15.84_{\pm0.26}$ |
| | Ens | $8.24_{\pm1.18}$ | $7.83_{\pm0.28}$ | $11.63_{\pm1.12}$ | $4.42_{\pm1.52}$ | $18.91_{\pm0.12}$ | $15.90_{\pm0.94}$ |
| | LA | $7.12_{\pm1.06}$ | $27.02_{\pm1.39}$ | $22.06_{\pm2.11}$ | $22.57_{\pm0.09}$ | $26.28_{\pm0.04}$ | $24.70_{\pm0.24}$ |
| | EDL | $\mathbf{8.38}_{\pm\mathbf{1.91}}$ | $26.16_{\pm4.44}$ | $35.64_{\pm2.97}$ | $44.29_{\pm2.42}$ | $46.47_{\pm0.24}$ | $30.59_{\pm1.14}$ |
| | VID | $19.14_{\pm0.99}$ | $\mathbf{13.28}_{\pm\mathbf{1.84}}$ | $\underline{15.00}_{\pm0.98}$ | $\underline{13.76}_{\pm0.90}$ | $\underline{23.49}_{\pm0.34}$ | $\underline{21.22}_{\pm0.63}$ |
| | I-EDL | $13.97_{\pm9.04}$ | $35.30_{\pm1.74}$ | $37.56_{\pm0.53}$ | $38.77_{\pm1.59}$ | $48.60_{\pm0.37}$ | $36.34_{\pm1.09}$ |
| | R-EDL | $12.25_{\pm1.18}$ | $20.67_{\pm2.30}$ | $28.02_{\pm2.63}$ | $33.64_{\pm0.29}$ | $39.58_{\pm0.56}$ | $30.42_{\pm0.89}$ |
| | IB-EDL | $\underline{11.21}_{\pm2.27}$ | $\underline{13.48}_{\pm1.61}$ | $\mathbf{12.30}_{\pm\mathbf{1.65}}$ | $\mathbf{10.48}_{\pm\mathbf{0.82}}$ | $\mathbf{23.47}_{\pm\mathbf{0.31}}$ | $\mathbf{21.15}_{\pm\mathbf{0.36}}$ |
| NLL ↓ | MAP | $1.94_{\pm0.06}$ | $1.05_{\pm0.09}$ | $0.74_{\pm0.03}$ | $0.84_{\pm0.02}$ | $0.50_{\pm0.01}$ | $0.73_{\pm0.00}$ |
| | MCD | $1.94_{\pm0.05}$ | $1.05_{\pm0.09}$ | $0.74_{\pm0.03}$ | $0.81_{\pm0.02}$ | $0.49_{\pm0.01}$ | $0.72_{\pm0.02}$ |
| | Ens | $1.37_{\pm0.21}$ | $0.73_{\pm0.06}$ | $0.73_{\pm0.03}$ | $\underline{0.83}_{\pm0.01}$ | $0.49_{\pm0.01}$ | $0.73_{\pm0.01}$ |
| | LA | $1.32_{\pm0.06}$ | $1.03_{\pm0.03}$ | $\underline{0.85}_{\pm0.00}$ | $0.92_{\pm0.01}$ | $0.56_{\pm0.00}$ | $0.81_{\pm0.00}$ |
| | EDL | $\underline{1.19}_{\pm0.03}$ | $0.97_{\pm0.03}$ | $1.05_{\pm0.02}$ | $1.35_{\pm0.11}$ | $0.92_{\pm0.01}$ | $0.90_{\pm0.03}$ |
| | VID | $1.27_{\pm0.01}$ | $0.83_{\pm0.03}$ | $\underline{0.76}_{\pm0.03}$ | $0.85_{\pm0.01}$ | $\underline{0.54}_{\pm0.01}$ | $\mathbf{0.71}_{\pm\mathbf{0.01}}$ |
| | I-EDL | $1.29_{\pm0.12}$ | $1.09_{\pm0.02}$ | $1.03_{\pm0.01}$ | $1.20_{\pm0.01}$ | $0.95_{\pm0.00}$ | $1.02_{\pm0.01}$ |
| | R-EDL | $1.24_{\pm0.01}$ | $0.92_{\pm0.01}$ | $0.95_{\pm0.03}$ | $1.13_{\pm0.01}$ | $0.83_{\pm0.01}$ | $0.93_{\pm0.01}$ |
| | IB-EDL | $\mathbf{1.18}_{\pm\mathbf{0.03}}$ | $\mathbf{0.74}_{\pm\mathbf{0.03}}$ | $\mathbf{0.71}_{\pm\mathbf{0.03}}$ | $\mathbf{0.81}_{\pm\mathbf{0.01}}$ | $\mathbf{0.52}_{\pm\mathbf{0.01}}$ | $\mathbf{0.71}_{\pm\mathbf{0.01}}$ |

## E.4  ADDITION CALIBRATION RESULTS ON OOD TEST SETS

In this section, we evaluate the calibration performance on OOD test sets to assess whether the uncertainty-aware methods can generalize effectively to OOD datasets. Specifically, we fine-tune Llama3-8B on the OBQA dataset and evaluate it on three OOD test sets. As shown in Table 12, IB-EDL achieves the best ECE and NLL on two out of the three OOD test sets, demonstrating that its calibration performance generalizes well to OOD datasets.

## E.5  ANALYSIS ON TRAINING AND INFERENCE SPEED, AND MEMORY CONSUMPTION

In this section, we evaluate the complexity of uncertainty-aware methods using three key metrics: (1) the number of samples processed per second during training; (2) the number of samples processed per second during inference; and (3) GPU memory consumption during training. For this experiment, we use Llama3-8B in mixed precision as the model and OBQA as both the training and test dataset. All experiments are conducted on a single NVIDIA H100 GPU. As shown in Table 13, IB-EDL demonstrates comparable training and inference speeds as well as similar memory consumption to MAP and other EDL baselines. Moreover, IB-EDL attains faster inference speeds compared to methods such as MCD and Ens, which require multiple forward passes, and LA, which involves gradient computation during inference. This substantial improvement in speed further highlights the computational efficiency of IB-EDL.

Table 9: Fine-tuning Llama3-8B on noisy datasets. In each training dataset, the labels of 30% samples are randomly perturbed. A robust uncertainty-aware method should not only maintain accuracy but also exhibit low calibration error. Non-EDL methods tend to significantly underperform EDL methods in terms of Accuracy. Therefore, the comparison for ECE and NLL is limited to EDL methods, with the best and second-best values among them highlighted.

| Metrics | Method | ARC-C | ARC-E | OBQA | CSQA | SciQ | RACE |
|---|---|---|---|---|---|---|---|
| Acc ↑ | MAP | $57.71_{\pm0.42}$ | $80.34_{\pm1.47}$ | $78.78_{\pm1.00}$ | $77.04_{\pm0.41}$ | $92.60_{\pm0.53}$ | $86.93_{\pm0.11}$ |
| | MCD | $57.71_{\pm0.42}$ | $80.37_{\pm1.46}$ | $79.00_{\pm0.92}$ | $77.05_{\pm0.41}$ | $92.87_{\pm0.12}$ | $86.93_{\pm0.12}$ |
| | Ens | $63.39_{\pm1.09}$ | $84.63_{\pm0.53}$ | $80.61_{\pm0.53}$ | $77.62_{\pm0.41}$ | $92.97_{\pm0.06}$ | $86.93_{\pm0.07}$ |
| | LA | $66.62_{\pm1.08}$ | $82.64_{\pm0.50}$ | $79.59_{\pm0.72}$ | $77.80_{\pm0.49}$ | $92.93_{\pm0.21}$ | $87.00_{\pm0.01}$ |
| | EDL | $\mathbf{69.43_{\pm0.98}}$ | $87.57_{\pm0.13}$ | $85.60_{\pm0.72}$ | $79.14_{\pm0.41}$ | $92.90_{\pm0.40}$ | $86.26_{\pm0.76}$ |
| | VID | $66.58_{\pm1.92}$ | $\underline{87.67_{\pm0.99}}$ | $84.86_{\pm1.01}$ | $\mathbf{79.66_{\pm1.26}}$ | $93.23_{\pm0.25}$ | $\underline{86.86_{\pm0.26}}$ |
| | I-EDL | $63.87_{\pm2.65}$ | $84.06_{\pm2.80}$ | $84.26_{\pm0.42}$ | $78.02_{\pm0.87}$ | $92.53_{\pm0.37}$ | $86.01_{\pm0.51}$ |
| | R-EDL | $71.25_{\pm1.20}$ | $84.91_{\pm3.91}$ | $\underline{85.73_{\pm0.70}}$ | $78.57_{\pm0.21}$ | $\mathbf{93.56_{\pm0.05}}$ | $86.16_{\pm0.42}$ |
| | IB-EDL | $\underline{68.53_{\pm0.25}}$ | $\mathbf{88.05_{\pm0.43}}$ | $\mathbf{86.13_{\pm0.51}}$ | $\underline{79.59_{\pm0.79}}$ | $\underline{93.46_{\pm0.38}}$ | $\mathbf{87.01_{\pm0.20}}$ |
| ECE ↓ | MAP | $13.26_{\pm0.85}$ | $8.52_{\pm0.63}$ | $3.72_{\pm1.06}$ | $8.43_{\pm1.04}$ | $16.29_{\pm0.43}$ | $19.61_{\pm0.25}$ |
| | MCD | $12.96_{\pm1.15}$ | $8.51_{\pm0.62}$ | $4.17_{\pm1.45}$ | $8.40_{\pm1.07}$ | $16.29_{\pm0.43}$ | $19.64_{\pm0.26}$ |
| | Ens | $10.41_{\pm0.74}$ | $11.09_{\pm0.59}$ | $4.24_{\pm1.91}$ | $8.21_{\pm0.21}$ | $16.15_{\pm0.16}$ | $19.69_{\pm0.23}$ |
| | LA | $19.02_{\pm0.78}$ | $23.63_{\pm1.74}$ | $11.61_{\pm0.29}$ | $18.05_{\pm0.86}$ | $18.36_{\pm0.37}$ | $20.99_{\pm0.18}$ |
| | EDL | $11.01_{\pm0.79}$ | $24.29_{\pm0.81}$ | $41.29_{\pm0.62}$ | $34.19_{\pm1.93}$ | $48.36_{\pm0.43}$ | $34.92_{\pm0.46}$ |
| | VID | $\mathbf{5.90_{\pm0.97}}$ | $\underline{19.58_{\pm0.29}}$ | $\underline{19.53_{\pm0.77}}$ | $\underline{15.22_{\pm1.15}}$ | $\underline{26.30_{\pm0.22}}$ | $\mathbf{22.25_{\pm0.25}}$ |
| | I-EDL | $9.57_{\pm2.10}$ | $33.57_{\pm6.96}$ | $42.64_{\pm0.36}$ | $41.89_{\pm1.18}$ | $50.13_{\pm0.31}$ | $42.49_{\pm0.20}$ |
| | R-EDL | $14.50_{\pm2.55}$ | $23.91_{\pm2.93}$ | $35.09_{\pm0.84}$ | $30.95_{\pm2.78}$ | $41.89_{\pm0.15}$ | $30.74_{\pm0.94}$ |
| | IB-EDL | $\underline{6.73_{\pm1.30}}$ | $\mathbf{19.49_{\pm0.38}}$ | $\mathbf{17.38_{\pm0.52}}$ | $\mathbf{11.73_{\pm1.11}}$ | $\mathbf{25.00_{\pm0.33}}$ | $\underline{23.43_{\pm0.32}}$ |
| NLL ↓ | MAP | $1.39_{\pm0.05}$ | $0.78_{\pm0.03}$ | $0.64_{\pm0.01}$ | $0.83_{\pm0.01}$ | $0.39_{\pm0.01}$ | $0.56_{\pm0.01}$ |
| | MCD | $1.38_{\pm0.02}$ | $0.78_{\pm0.03}$ | $0.64_{\pm0.03}$ | $0.82_{\pm0.02}$ | $0.40_{\pm0.01}$ | $0.57_{\pm0.01}$ |
| | Ens | $1.15_{\pm0.05}$ | $0.64_{\pm0.03}$ | $0.61_{\pm0.01}$ | $0.78_{\pm0.02}$ | $0.39_{\pm0.01}$ | $0.57_{\pm0.02}$ |
| | LA | $1.11_{\pm0.01}$ | $0.80_{\pm0.02}$ | $0.63_{\pm0.02}$ | $0.85_{\pm0.01}$ | $0.39_{\pm0.00}$ | $0.57_{\pm0.01}$ |
| | EDL | $\mathbf{0.94_{\pm0.02}}$ | $0.70_{\pm0.01}$ | $0.92_{\pm0.01}$ | $1.04_{\pm0.03}$ | $0.85_{\pm0.00}$ | $0.78_{\pm0.01}$ |
| | VID | $0.98_{\pm0.03}$ | $\underline{0.62_{\pm0.02}}$ | $\underline{0.63_{\pm0.01}}$ | $\mathbf{0.80_{\pm0.02}}$ | $\underline{0.49_{\pm0.01}}$ | $\mathbf{0.57_{\pm0.01}}$ |
| | I-EDL | $1.03_{\pm0.03}$ | $0.87_{\pm0.06}$ | $0.96_{\pm0.01}$ | $1.16_{\pm0.00}$ | $0.89_{\pm0.01}$ | $0.91_{\pm0.00}$ |
| | R-EDL | $1.00_{\pm0.02}$ | $0.76_{\pm0.05}$ | $0.83_{\pm0.02}$ | $0.99_{\pm0.02}$ | $0.72_{\pm0.00}$ | $0.73_{\pm0.01}$ |
| | IB-EDL | $\underline{0.99_{\pm0.02}}$ | $\mathbf{0.60_{\pm0.01}}$ | $\mathbf{0.57_{\pm0.01}}$ | $\underline{0.74_{\pm0.02}}$ | $\mathbf{0.46_{\pm0.01}}$ | $\mathbf{0.57_{\pm0.01}}$ |

### E.6 SENSITIVITY ANALYSIS ON THE NUMBER OF BINS OF ECE

Following Yang et al. (2024), we use 15 bins by default when measuring ECE. To assess the impact of this hyperparameter, we conduct a sensitivity analysis by varying the number of bins across $\{10, 15, 25, 35\}$. For this experiment, we train and test Llama3-8B on the OBQA dataset and calculate the ECE for each bin setting. As shown in Table 14, although ECE values increase slightly with a higher number of bins, the relative rankings of the methods remain largely consistent.

Table 10: Fine-tuning Mistral-7B on noisy datasets. In each training dataset, the labels of 30% samples are randomly perturbed. A robust uncertainty-aware method should not only maintain accuracy but also exhibit low calibration error. Non-EDL methods tend to significantly underperform EDL methods in terms of Accuracy. Therefore, the comparison for ECE and NLL is limited to EDL methods, with the best and second-best values among them highlighted.

| Metrics | Method | ARC-C | ARC-E | OBQA | CSQA | SciQ | RACE |
|---|---|---|---|---|---|---|---|
| Acc ↑ | MAP | $50.65_{\pm0.30}$ | $66.30_{\pm2.16}$ | $75.72_{\pm0.46}$ | $69.81_{\pm0.55}$ | $92.96_{\pm0.32}$ | $85.47_{\pm0.35}$ |
| | MCD | $50.65_{\pm0.30}$ | $66.21_{\pm2.04}$ | $75.46_{\pm0.24}$ | $69.77_{\pm0.51}$ | $92.93_{\pm0.31}$ | $85.46_{\pm0.35}$ |
| | Ens | $58.16_{\pm0.47}$ | $76.10_{\pm0.86}$ | $76.93_{\pm2.14}$ | $74.18_{\pm1.50}$ | $92.93_{\pm0.39}$ | $85.62_{\pm0.28}$ |
| | LA | $66.35_{\pm0.63}$ | $75.48_{\pm1.88}$ | $78.20_{\pm1.25}$ | $74.24_{\pm1.01}$ | $93.02_{\pm0.29}$ | $85.57_{\pm0.06}$ |
| | EDL | $61.63_{\pm1.80}$ | $85.01_{\pm0.23}$ | $83.20_{\pm1.25}$ | $76.63_{\pm0.98}$ | $93.33_{\pm0.06}$ | $84.91_{\pm0.10}$ |
| | VID | $62.27_{\pm2.18}$ | $76.98_{\pm1.13}$ | $82.06_{\pm1.51}$ | $77.31_{\pm1.01}$ | $93.20_{\pm0.17}$ | $\underline{85.59_{\pm0.06}}$ |
| | I-EDL | $\underline{68.28_{\pm1.19}}$ | $79.73_{\pm0.57}$ | $81.06_{\pm0.81}$ | $\underline{77.41_{\pm0.09}}$ | $93.33_{\pm0.31}$ | $84.44_{\pm0.28}$ |
| | R-EDL | $\mathbf{76.39_{\pm0.60}}$ | $\mathbf{86.85_{\pm0.69}}$ | $82.76_{\pm1.73}$ | $77.10_{\pm1.10}$ | $93.09_{\pm0.78}$ | $85.07_{\pm0.34}$ |
| | IB-EDL | $66.44_{\pm0.80}$ | $85.17_{\pm1.39}$ | $\mathbf{84.33_{\pm0.51}}$ | $\mathbf{77.44_{\pm0.39}}$ | $\mathbf{93.63_{\pm0.21}}$ | $\mathbf{85.68_{\pm0.20}}$ |
| ECE ↓ | MAP | $18.08_{\pm0.45}$ | $9.95_{\pm1.18}$ | $7.83_{\pm0.62}$ | $9.61_{\pm0.98}$ | $15.11_{\pm0.28}$ | $17.80_{\pm0.33}$ |
| | MCD | $18.24_{\pm0.65}$ | $9.94_{\pm1.18}$ | $7.82_{\pm0.61}$ | $9.60_{\pm0.99}$ | $15.10_{\pm0.27}$ | $17.80_{\pm0.33}$ |
| | Ens | $10.94_{\pm0.21}$ | $12.40_{\pm0.34}$ | $7.02_{\pm0.61}$ | $8.72_{\pm0.29}$ | $15.09_{\pm0.17}$ | $18.58_{\pm0.46}$ |
| | LA | $18.50_{\pm1.46}$ | $17.31_{\pm0.23}$ | $13.92_{\pm1.45}$ | $15.04_{\pm0.41}$ | $18.13_{\pm0.19}$ | $18.32_{\pm0.59}$ |
| | EDL | $12.76_{\pm0.41}$ | $19.84_{\pm0.95}$ | $32.57_{\pm1.89}$ | $25.26_{\pm5.63}$ | $48.55_{\pm0.33}$ | $34.06_{\pm0.14}$ |
| | VID | $\mathbf{6.27_{\pm0.66}}$ | $\mathbf{13.17_{\pm0.76}}$ | $\mathbf{13.92_{\pm0.97}}$ | $\underline{11.66_{\pm1.17}}$ | $\underline{26.45_{\pm0.16}}$ | $\mathbf{22.01_{\pm0.08}}$ |
| | I-EDL | $13.14_{\pm1.17}$ | $21.44_{\pm1.54}$ | $35.10_{\pm1.07}$ | $33.29_{\pm1.04}$ | $50.90_{\pm0.44}$ | $41.19_{\pm0.27}$ |
| | R-EDL | $17.63_{\pm2.83}$ | $\underline{17.57_{\pm0.42}}$ | $18.55_{\pm1.68}$ | $21.35_{\pm1.24}$ | $41.48_{\pm0.26}$ | $29.40_{\pm0.31}$ |
| | IB-EDL | $\underline{7.46_{\pm1.58}}$ | $19.08_{\pm3.68}$ | $\underline{14.90_{\pm0.50}}$ | $\mathbf{11.58_{\pm0.43}}$ | $\mathbf{25.08_{\pm0.27}}$ | $\underline{22.04_{\pm0.06}}$ |
| NLL ↓ | MAP | $1.67_{\pm0.03}$ | $0.98_{\pm0.06}$ | $0.73_{\pm0.05}$ | $0.98_{\pm0.03}$ | $0.38_{\pm0.01}$ | $0.56_{\pm0.00}$ |
| | MCD | $1.67_{\pm0.03}$ | $0.98_{\pm0.06}$ | $0.70_{\pm0.03}$ | $0.98_{\pm0.03}$ | $0.38_{\pm0.01}$ | $0.56_{\pm0.01}$ |
| | Ens | $1.32_{\pm0.01}$ | $0.73_{\pm0.03}$ | $0.64_{\pm0.04}$ | $0.81_{\pm0.02}$ | $0.39_{\pm0.01}$ | $0.57_{\pm0.02}$ |
| | LA | $1.08_{\pm0.03}$ | $0.83_{\pm0.03}$ | $0.70_{\pm0.03}$ | $0.93_{\pm0.01}$ | $0.40_{\pm0.01}$ | $0.56_{\pm0.00}$ |
| | EDL | $\mathbf{0.93_{\pm0.05}}$ | $\underline{0.67_{\pm0.00}}$ | $0.82_{\pm0.03}$ | $0.99_{\pm0.04}$ | $0.85_{\pm0.00}$ | $0.78_{\pm0.01}$ |
| | VID | $1.04_{\pm0.06}$ | $0.74_{\pm0.02}$ | $\underline{0.63_{\pm0.01}}$ | $\underline{0.84_{\pm0.01}}$ | $\underline{0.50_{\pm0.01}}$ | $\mathbf{0.60_{\pm0.01}}$ |
| | I-EDL | $0.96_{\pm0.04}$ | $0.76_{\pm0.02}$ | $0.90_{\pm0.01}$ | $1.05_{\pm0.02}$ | $0.89_{\pm0.00}$ | $0.91_{\pm0.01}$ |
| | R-EDL | $\mathbf{0.93_{\pm0.03}}$ | $\mathbf{0.64_{\pm0.02}}$ | $0.64_{\pm0.03}$ | $0.97_{\pm0.04}$ | $0.72_{\pm0.01}$ | $0.72_{\pm0.01}$ |
| | IB-EDL | $0.98_{\pm0.02}$ | $0.69_{\pm0.03}$ | $\mathbf{0.57_{\pm0.01}}$ | $\mathbf{0.81_{\pm0.01}}$ | $\mathbf{0.45_{\pm0.01}}$ | $\mathbf{0.60_{\pm0.01}}$ |

Table 11: OOD Detection AUROC on Llama3-8B in the setting OBQA → MMLU-Math. IB-EDL achieves the best performance even under significant distribution shifts, such as transitioning from a common-sense reasoning dataset like OBQA to a math-focused OOD dataset.

| Method | OBQA → MMLU-Math | |
|---|---|---|
| | AUROC ↑ | |
| | MP | UM |
| MAP | $91.36_{\pm0.57}$ | - |
| MCD | $90.85_{\pm0.33}$ | - |
| Ens | $90.68_{\pm0.80}$ | - |
| LA | $91.09_{\pm0.41}$ | - |
| EDL | $92.78_{\pm0.26}$ | $92.86_{\pm0.21}$ |
| VID | $91.64_{\pm0.79}$ | $66.61_{\pm4.98}$ |
| I-EDL | $91.48_{\pm0.72}$ | $90.67_{\pm0.88}$ |
| R-EDL | $88.44_{\pm2.11}$ | $88.22_{\pm1.70}$ |
| IB-EDL | $\mathbf{93.63_{\pm0.66}}$ | $\mathbf{93.64_{\pm0.56}}$ |

Table 12: Calibration performance of uncertainty-aware methods on fine-tuned Llama3-8B in the OOD setting. The model is trained on OBQA and tested on three different OOD test sets.

| Metrics | Method | OBQA → ARC-C | OBQA → ARC-E | OBQA → CSQA |
|---------|--------|--------------|--------------|-------------|
| Acc ↑ | MAP | $79.18_{\pm0.45}$ | $88.06_{\pm0.20}$ | $69.37_{\pm0.67}$ |
| | MCD | $79.16_{\pm0.43}$ | $88.05_{\pm0.20}$ | $69.38_{\pm0.68}$ |
| | Ens | $\underline{79.27}_{\pm0.25}$ | $\underline{88.15}_{\pm0.02}$ | $69.14_{\pm0.47}$ |
| | LA | $\mathbf{79.38}_{\pm\mathbf{0.40}}$ | $\mathbf{88.36}_{\pm\mathbf{0.24}}$ | $69.34_{\pm0.58}$ |
| | EDL | $78.27_{\pm0.79}$ | $86.38_{\pm0.87}$ | $69.34_{\pm0.88}$ |
| | VID | $78.27_{\pm0.57}$ | $87.41_{\pm0.82}$ | $69.99_{\pm1.07}$ |
| | I-EDL | $78.55_{\pm0.30}$ | $87.55_{\pm0.20}$ | $70.49_{\pm0.56}$ |
| | R-EDL | $78.32_{\pm1.24}$ | $87.31_{\pm0.93}$ | $\underline{70.62}_{\pm1.28}$ |
| | IB-EDL | $78.31_{\pm1.14}$ | $87.94_{\pm0.22}$ | $\mathbf{71.29}_{\pm\mathbf{0.96}}$ |
| ECE ↓ | MAP | $18.04_{\pm0.38}$ | $9.63_{\pm0.47}$ | $27.85_{\pm0.53}$ |
| | MCD | $18.06_{\pm0.36}$ | $9.63_{\pm0.46}$ | $27.86_{\pm0.46}$ |
| | Ens | $16.74_{\pm0.13}$ | $9.19_{\pm0.65}$ | $27.07_{\pm1.32}$ |
| | LA | $6.61_{\pm0.41}$ | $\mathbf{3.02}_{\pm\mathbf{0.15}}$ | $11.99_{\pm0.72}$ |
| | EDL | $7.65_{\pm0.47}$ | $10.11_{\pm0.85}$ | $8.32_{\pm1.33}$ |
| | VID | $8.74_{\pm1.19}$ | $\underline{3.19}_{\pm0.19}$ | $16.66_{\pm0.73}$ |
| | I-EDL | $7.68_{\pm1.19}$ | $11.38_{\pm0.65}$ | $\underline{5.96}_{\pm0.71}$ |
| | R-EDL | $\underline{5.03}_{\pm0.62}$ | $5.32_{\pm0.46}$ | $12.84_{\pm1.73}$ |
| | IB-EDL | $\mathbf{4.67}_{\pm\mathbf{1.09}}$ | $5.03_{\pm0.15}$ | $\mathbf{4.51}_{\pm\mathbf{0.15}}$ |
| NLL ↓ | MAP | $1.30_{\pm0.02}$ | $0.71_{\pm0.03}$ | $2.06_{\pm0.04}$ |
| | MCD | $1.33_{\pm0.06}$ | $0.70_{\pm0.04}$ | $2.13_{\pm0.13}$ |
| | Ens | $1.19_{\pm0.03}$ | $0.68_{\pm0.05}$ | $2.01_{\pm0.10}$ |
| | LA | $0.73_{\pm0.02}$ | $\mathbf{0.42}_{\pm\mathbf{0.02}}$ | $1.15_{\pm0.01}$ |
| | EDL | $0.74_{\pm0.01}$ | $0.52_{\pm0.01}$ | $0.98_{\pm0.03}$ |
| | VID | $0.78_{\pm0.01}$ | $0.46_{\pm0.02}$ | $1.06_{\pm0.02}$ |
| | I-EDL | $0.73_{\pm0.02}$ | $0.53_{\pm0.01}$ | $\underline{0.94}_{\pm0.01}$ |
| | R-EDL | $\underline{0.73}_{\pm0.04}$ | $0.46_{\pm0.01}$ | $1.00_{\pm0.05}$ |
| | IB-EDL | $\mathbf{0.72}_{\pm\mathbf{0.02}}$ | $\underline{0.44}_{\pm0.02}$ | $\mathbf{0.93}_{\pm\mathbf{0.02}}$ |

Table 13: Comparison of computational efficiency for various methods using Llama3-8B on the OBQA. IB-EDL demonstrates comparable training and inference speeds as well as memory consumption compared to MAP and other EDL methods, confirming its computational efficiency.

| Method | Test Samples/s ↑ | Training Samples/s ↑ | Memory (GB) at Training↓ |
|--------|------------------|----------------------|--------------------------|
| MAP | $69.55_{\pm2.86}$ | $26.57_{\pm1.96}$ | $21.21_{\pm0.35}$ |
| MCD (10 forwards) | $9.79_{\pm1.21}$ | - | - |
| Ens (3 models) | $25.77_{\pm3.54}$ | - | - |
| LA | $5.95_{\pm0.49}$ | - | - |
| EDL | $68.99_{\pm1.59}$ | $26.44_{\pm1.11}$ | $21.23_{\pm0.15}$ |
| VID | $69.17_{\pm0.99}$ | $26.69_{\pm1.37}$ | $21.29_{\pm0.37}$ |
| I-EDL | $68.94_{\pm2.18}$ | $26.02_{\pm0.71}$ | $21.33_{\pm0.11}$ |
| R-EDL | $68.84_{\pm1.09}$ | $26.47_{\pm1.09}$ | $21.27_{\pm0.21}$ |
| IB-EDL | $68.08_{\pm1.75}$ | $26.41_{\pm1.04}$ | $21.88_{\pm0.66}$ |

Table 14: Sensitivity analysis of ECE with respect to the number of bins. We train and test Llama3-8B on the OBQA dataset. The results show that while the ECE values increase slightly as the number of bins increases, the relative rankings of the methods remain consistent.

| Method | Bins = 10 | Bins = 15 | Bins = 25 | Bins = 35 |
|--------|-----------|-----------|-----------|-----------|
| MAP | $10.45_{\pm 0.52}$ | $10.52_{\pm 0.87}$ | $10.89_{\pm 0.67}$ | $10.99_{\pm 1.01}$ |
| MCD | $10.31_{\pm 0.37}$ | $10.48_{\pm 0.86}$ | $10.69_{\pm 0.59}$ | $10.83_{\pm 0.76}$ |
| Ens | $10.11_{\pm 0.13}$ | $10.08_{\pm 0.90}$ | $10.91_{\pm 0.40}$ | $10.92_{\pm 0.84}$ |
| LA | $5.20_{\pm 1.29}$ | $5.26_{\pm 1.30}$ | $6.33_{\pm 1.11}$ | $6.42_{\pm 0.96}$ |
| EDL | $8.16_{\pm 1.25}$ | $8.28_{\pm 1.62}$ | $8.78_{\pm 1.27}$ | $9.44_{\pm 1.79}$ |
| VID | $5.29_{\pm 0.50}$ | $5.99_{\pm 1.41}$ | $7.16_{\pm 1.59}$ | $7.34_{\pm 1.17}$ |
| I-EDL | $7.31_{\pm 0.33}$ | $7.57_{\pm 0.52}$ | $8.20_{\pm 0.46}$ | $9.01_{\pm 0.50}$ |
| R-EDL | $4.64_{\pm 0.87}$ | $4.68_{\pm 1.35}$ | $4.81_{\pm 1.09}$ | $5.47_{\pm 0.82}$ |
| IB-EDL | $\mathbf{2.77_{\pm 0.61}}$ | $\mathbf{2.34_{\pm 0.61}}$ | $\mathbf{3.91_{\pm 0.77}}$ | $\mathbf{4.54_{\pm 0.52}}$ |

