# OpenReview forum: "Calibrating LLMs with Information-Theoretic Evidential Deep Learning"
_ICLR.cc/2025/Conference — ICLR 2025 Poster_

### Official Review · Reviewer_4Vtx · 2024-11-04

**Soundness:** 3
**Presentation:** 3
**Contribution:** 2
**Rating:** 6
**Confidence:** 4

**Summary:**

The paper presents the Information Bottleneck (IB) - Evidential Deep Learning (EDL) method, designed to regularize EDL for better calibration of large language models (LLMs). The empirical experimental results demonstrate that IB-EDL enhances calibration in fine-tuned LLMs.

**Strengths:**

The paper is well-written and easy to understand, trying to address the important task of calibrating fine-tuned large language models. The empirical experiments are comprehensive, testing across multiple LLMs including Llama2-7B, Llama3-8B, and Mistral-7B. The paper covers most of the baselines. The LoRA-based Mixture of Experts idea [1] could also be relevant and serve as a baseline for comparison. The empirical performance appears promising.

[1] Li, Dengchun, et al. "Mixlora: Enhancing large language models fine-tuning with lora based mixture of experts."

**Weaknesses:**

The novelty of the paper is limited, as both EDL and the IB method are well studied in the literature. The primary contribution of the paper appears to be the implement IB on EDL. The authors explain that the IB objective allows the latent variables to retain the most predictive information about the target variable while discarding irrelevant information from X. However, a more detailed discussion on how this relates to improved calibration of LLMs is missing. Additionally, there is a lack of theoretical guarantees supporting this approach. It is beneficial to provide a theoretical bound on calibration error or a proof of convergence.

In terms of the empirical analysis, the paper highlights computational efficiency as a key contribution of the proposed method, yet it lacks experimental comparisons of computational time and memory costs to validating this claim. The paper also lacks the calibration performance analysis in out-of-distribution scenarios.

**Questions:**

Please refer to the strengths and weaknesses. Additionally, how many bins did the authors use to compute the ECE? It is beneficial to conduct a sensitivity analysis on the number of bins.

---

> ### Author Response · Authors · 2024-11-17
> **Response to Reviewer 4Vtx (1/3)**
>
> We thank the reviewer for the constructive comments and feedback. We address the concerns as follows:
>
> >S1: The LoRA-based Mixture of Experts idea [1] could also be relevant.
>
> We appreciate the reviewer for bringing this work to our attention. We will include it in the Related Work in a revised version of the manuscript, which we will upload later during the discussion phase.
>
> >W1: The novelty of the paper is limited, as both EDL and the IB method are well studied in the literature. The primary contribution of the paper appears to be the implement IB on EDL.
>
> Our work is not a simple combination of these methods; instead, **it tackles key unresolved issues in EDL (see also [2, 3]) and addresses the emerged challenge when regularizing EDL with IB**. Importantly, implementing IB on EDL in a naïve manner, such as inserting IB at an arbitrary layer, can compromise theoretical soundness. Our contributions go beyond mere implementation and address these challenges through the following advancements:
>
> 1. **Uncovering theoretical limitations:** We identify a fundamental issue overlooked in prior literature [4, 5]: applying IB at arbitrary layers in an LLM can result in an optimization objective that no longer serves as a valid lower bound of $I(Z, Y)$. In Section 3.1, we provide a detailed theoretical analysis (Eq. 11) that highlights this limitation.
>
> 2. **Proposing a novel solution:** In Section 3.2, we propose a unique solution by selecting the pre-evidence $\tilde{e}$ as the IB variable. This choice ensures that the optimization objective remains theoretically sound. Furthermore, as shown in Eq. (14), our method introduces L2 regularization on evidence values, reducing overconfidence by penalizing extreme outputs. This directly addresses critical issues in EDL identified in prior works [2, 3].
>
> 3. **Unifying prior approaches:** Through Proposition 1, we provide a cohesive framework that unifies prior approaches in EDL, offering deeper insights and a clearer understanding of their relationship to IB. This unified perspective is novel.
>
> >W2: The authors explain that the IB objective allows the latent variables to retain the most predictive information about the target variable while discarding irrelevant information from X. However, a more detailed discussion on how this relates to improved calibration of LLMs is missing.
>
> Overparameterized LLMs often exhibit overconfidence when fine-tuned on small datasets or limited human feedback, primarily due to overfitting to spurious correlations in the training data. We utilize IB to address this issue by encouraging the model to focus on truly predictive features and discard irrelevant or spurious correlations. This, in turn, helps the model to learn more generalizable features and reduces overconfident predictions, leading to improved calibration. We will make the connection more clear in a revised manuscript version and thank the reviewer for pointing this out.
>
> >W3: Additionally, there is a lack of theoretical guarantees supporting this approach. It is beneficial to provide a theoretical bound on calibration error or a proof of convergence.
>
> 1. **Theoretical guarantees in our work**: We provide several theoretical guarantees: (i) Through our analysis in Eq. (11) and Section 3.2, we ensure that our optimization objective remains a valid lower bound of $I(Y, Z)$; (ii) Through Eq. (14), we show that our approach penalizes overconfident predictions by introducing L2 regularization on evidence, preventing extreme values.
>
> 2. **On calibration error bounds**: Although deriving a formal bound on calibration error would further strengthen our contributions, it requires further analysis and is beyond the scope of this work.  In particular, theoretical bounds for calibration error have not been studied in EDL or IB literature so far.  Nonetheless, using Theorem 2 from [6], we can at least infer a **generalization bound** for IB-EDL. For any $\delta > 0$, the theorem guarantees that with probability at least $1 - \delta$, the generalization bound $\Delta$ for $n$ data points is $\Delta(n) \leq \min_l Q_l$, where $Q_l$ is a bounded term dependent on the $l$-th layer and number of observations $n$. The theorem by [6] is the tightest guarantee currently known with linear growth rates depending on $I(X,Z|Y)$, which is indirectly minimized by our objective. So we could apply the convexity of $\min$ function to derive a generalization bound $\Delta(n) \leq Q_L$, where $L$ is the layer index of the pre-evidence $\tilde{e}$ used in IB-EDL.

---

> ### Author Response · Authors · 2024-11-17
> **Response to Reviewer 4Vtx (2/3)**
>
> >W4: The paper highlights computational efficiency as a key contribution of the proposed method, yet it lacks experimental comparisons of computational time and memory costs to validating this claim
>
> In Section 4.5, we provided a comparison of FLOPs and model parameters. Here, we offer a more comprehensive analysis, including training and inference time, as well as memory usage. Specifically, we evaluate IB-EDL using Llama3-8B on the OBQA dataset with a single NVIDIA H100 GPU. As shown in Table A, **IB-EDL’s training and inference speeds, as well as its memory consumption, are comparable to MAP and other EDL methods**. This confirms the computational efficiency of our approach.
>
> ### Table A: Additional Complexity Analysis
> | Method            | Test Samples/s | Training Samples/s | Training Time Memory (GB) |
> |-------------------|----------------|--------------------|---------------------------|
> | MAP               | 69.55 ± 2.86   | 26.57 ± 1.96       | 21.21 ± 0.35              |
> | MCD (10 forwards) | 9.79 ± 1.21    | -                  | -                         |
> | Ens (3 models)    | 25.77 ± 3.54   | -                  | -                         |
> | LA                | 5.95 ± 0.49    | -                  | -                         |
> | EDL               | 68.99 ± 1.59   | 26.44 ± 1.11       | 21.23 ± 0.15              |
> | VID               | 69.17 ± 0.99   | 26.69 ± 1.37       | 21.29 ± 0.37              |
> | I-EDL             | 68.94 ± 2.18   | 26.02 ± 0.71       | 21.33 ± 0.11              |
> | R-EDL             | 68.84 ± 1.09   | 26.47 ± 1.09       | 21.27 ± 0.21              |
> | IB-EDL  (Ours)     | 68.08 ± 1.75   | 26.41 ± 1.04       | 21.88 ± 0.66              |
>
> >W5: The paper also lacks the calibration performance analysis in out-of-distribution scenarios.
>
> Thank you for raising this point. We provide calibration results on OOD datasets in Tables B, C, and D. These results demonstrate that IB-EDL achieves the best ECE and NLL on two out of three OOD datasets. This indicates that **IB-EDL’s calibration abilities generalize well to distribution shifts**, further validating its robustness.
>
> ### Table B: Calibration in the setting OBQA -> ARC-C
> | Method | Acc              | ECE             | NLL             |
> |--------|------------------|-----------------|-----------------|
> | MAP    | 79.18 ± 0.45     | 18.04 ± 0.38    | 1.30 ± 0.02     |
> | MCD    | 79.16 ± 0.43     | 18.06 ± 0.36    | 1.33 ± 0.06     |
> | Ens    | 79.27 ± 0.45     | 16.74 ± 0.13    | 1.19 ± 0.03     |
> | LA     | **79.38 ± 0.40** | 6.61 ± 0.41     | 0.73 ± 0.02     |
> | EDL    | 78.27 ± 0.79     | 7.65 ± 0.47     | 0.74 ± 0.01     |
> | VID    | 78.27 ± 0.57     | 8.74 ± 0.40     | 0.78 ± 0.01     |
> | I-EDL  | 78.55 ± 0.30     | 7.68 ± 1.19     | 0.73 ± 0.02     |
> | R-EDL  | 78.32 ± 1.24     | 5.03 ± 0.62     | 0.73 ± 0.04     |
> | IB-EDL (Ours) | 78.31 ± 1.14     | **4.67 ± 1.09** | **0.72 ± 0.02** |
>
> ### Table C: Calibration in the setting OBQA -> ARC-E
> | Method | Acc              | ECE             | NLL             |
> |--------|------------------|-----------------|-----------------|
> | MAP    | 88.06 ± 0.20     | 9.63 ± 0.47     | 0.71 ± 0.03     |
> | MCD    | 88.05 ± 0.20     | 9.63 ± 0.46     | 0.70 ± 0.04     |
> | Ens    | 88.15 ± 0.02     | 9.19 ± 0.65     | 0.68 ± 0.05     |
> | LA     | **88.36 ± 0.24** | **3.02 ± 0.15** | **0.42 ± 0.02** |
> | EDL    | 86.38 ± 0.87     | 10.11 ± 0.85    | 0.52 ± 0.01     |
> | VID    | 87.41 ± 0.82     | 3.19 ± 0.19     | 0.46 ± 0.02     |
> | I-EDL  | 87.55 ± 0.20     | 11.83 ± 0.65    | 0.53 ± 0.01     |
> | R-EDL  | 87.31 ± 0.93     | 5.32 ± 0.46     | 0.46 ± 0.01     |
> | IB-EDL (Ours) | 87.94 ± 0.22     | 5.03 ± 0.15     | 0.44 ± 0.02     |
>
> ### Table D: Calibration in the setting OBQA -> CSQA
> | Method | Acc              | ECE             | NLL             |
> |--------|------------------|-----------------|-----------------|
> | MAP    | 69.37 ± 0.67     | 27.85 ± 0.53    | 2.06 ± 0.04     |
> | MCD    | 69.38 ± 0.68     | 27.86 ± 0.46    | 2.13 ± 0.13     |
> | Ens    | 69.14 ± 0.47     | 27.07 ± 1.32    | 2.01 ± 0.10     |
> | LA     | 69.34 ± 0.58     | 11.99 ± 0.72    | 1.15 ± 0.01     |
> | EDL    | 69.34 ± 0.88     | 8.32 ± 1.33     | 0.98 ± 0.03     |
> | VID    | 69.99 ± 1.07     | 16.66 ± 0.73    | 1.06 ± 0.02     |
> | I-EDL  | 70.49 ± 0.56     | 5.96 ± 0.71     | 0.94 ± 0.01     |
> | R-EDL  | 70.62 ± 1.28     | 12.84 ± 1.73    | 1.00 ± 0.05     |
> | IB-EDL (Ours) | **71.29 ± 0.96** | **4.51 ± 0.15** | **0.93 ± 0.02** |

---

> ### Author Response · Authors · 2024-11-17
> **Response to Reviewer 4Vtx (3/3)**
>
> >Q1: Additionally, how many bins did the authors use to compute the ECE? It is beneficial to conduct a sensitivity analysis on the number of bins.
>
> Following the implementation of LA [7], we use n_bins = 15 by default for computing ECE. To address this concern, we provide a sensitivity analysis using Llama3-8B on OBQA, as shown in Table F. The results show that **while ECE values increase slightly as the number of bins increases, the relative rankings of the methods basically remain consistent**.
>
> ### Table F: Sensitivity Analysis of the Number of Bins
> | Method | Bins = 10       | Bins = 15       | Bins = 25       | Bins = 35       |
> |--------|-----------------|-----------------|-----------------|-----------------|
> | MAP    | 10.45 ± 0.52    | 10.52 ± 0.87    | 10.89 ± 0.67    | 10.99 ± 1.01    |
> | MCD    | 10.31 ± 0.37    | 10.48 ± 0.86    | 10.69 ± 0.59    | 10.83 ± 0.76    |
> | Ens    | 10.11 ± 0.13    | 10.08 ± 0.90    | 10.91 ± 0.40    | 10.92 ± 0.84    |
> | LA     | 5.20 ± 1.29     | 5.26 ± 1.30     | 6.33 ± 1.11     | 6.42 ± 0.96     |
> | EDL    | 8.16 ± 1.25     | 8.28 ± 1.62     | 8.78 ± 1.27     | 9.44 ± 1.79     |
> | VID    | 5.29 ± 0.50     | 5.99 ± 1.41     | 7.16 ± 1.59     | 7.34 ± 1.17     |
> | I-EDL  | 7.31 ± 0.33     | 7.57 ± 0.52     | 8.20 ± 0.46     | 9.01 ± 0.50     |
> | R-EDL  | 4.64 ± 0.87     | 4.68 ± 1.35     | 4.81 ± 1.09     | 5.47 ± 0.82     |
> | IB-EDL (Ours) | **2.77 ± 0.61** | **2.34 ± 0.61** | **3.91 ± 0.77** | **4.54 ± 0.52** |
>
> —
>
> Thank you again for your constructive feedback. Please let us know if our response addresses your concerns, and if there is anything else we need to clarify.
>
>
> [2] Deng, Danruo, et al. "Uncertainty estimation by fisher information-based evidential deep learning." International Conference on Machine Learning. PMLR, 2023.
>
> [3] Chen, Mengyuan, Junyu Gao, and Changsheng Xu. "R-EDL: Relaxing Nonessential Settings of Evidential Deep Learning." The Twelfth International Conference on Learning Representations.
>
> [4] Alemi, Alexander A., et al. "Deep variational information bottleneck." arXiv preprint arXiv:1612.00410 (2016).
>
> [5] Wieczorek, Aleksander, and Volker Roth. "On the difference between the information bottleneck and the deep information bottleneck." Entropy 22.2 (2020): 131.
>
> [6] Kawaguchi, Kenji, et al. "How does information bottleneck help deep learning?." International Conference on Machine Learning. PMLR, 2023.
>
> [7] Yang, Adam X., et al. "Bayesian Low-rank Adaptation for Large Language Models." The Twelfth International Conference on Learning Representations.

---

> ### Author Response · Authors · 2024-11-24
> **The end of the rebuttal phase is approaching**
>
> Dear Reviewer 4Vtx,
>
> Thank you for providing valuable feedback on our manuscript. We greatly appreciate your recognition of our effort in the important task of calibrating LLMs  and our strong experiment results.
>
> As the discussion period is set to conclude in approximately 72 hours, we wanted to check if our responses thus far have sufficiently addressed your concerns and clarified your questions.
>
> In summary, we addressed the points raised by you as follows:
>
> * **Incorporating relevant related work:** We have added MixLoRA in the Related Work section of a revised manuscript.
>
> * **Novelty of our contributions:** We clarified that our method goes beyond a naive implementation of IB on EDL, we uncovered theoretical challenges overlooked by previous IB literature (Section 3.1), and we address these challenges through a novel IB variable parameterization to maintain theoretical soundness (Section 3.2).
>
> * **Connection between IB and LLM calibration:** We elaborated on why EDL is suitable for calibrating LLMs and why IB can further enhance the calibration of EDL methods.
>
> * **Theoretical guarantees:** While deriving explicit bounds on ECE is beyond the scope of our work, we discussed existing theoretical guarantees related to generalization bounds that indirectly support our approach.
>
> * **Computational efficiency:** We provided additional analyses of training and inference time, as well as memory usage (Table A), which demonstrate that IB-EDL’s computational requirements are comparable to MAP and other EDL baselines.
>
> * **Calibration in OOD scenarios:** We addressed this concern by presenting additional experiments on OOD datasets (Tables B, C, and D), showing that IB-EDL achieves superior calibration performance under distribution shifts.
>
> * **Sensitivity to bin count in ECE:** We conducted a sensitivity analysis on the number of bins for ECE calculation (Table F), showing that the rankings of methods remain consistent across different configurations.
>
> If there are any remaining questions or concerns, please let us know.
>
> Should you find that our responses sufficiently address your feedback, we would sincerely appreciate it if you consider raising the score.
>
> Best regards,
>
> Authors of IB-EDL

---

> > ### Comment · Reviewer_4Vtx · 2024-11-27
> > **Response**
> >
> > Thanks for the response. I have raised the score to 6.

---

> > > ### Author Response · Authors · 2024-11-27
> > > **Thank you**
> > >
> > > Dear Reviewer 4Vtx,
> > >
> > > Thank you for your thoughtful review and for taking the time to assess our (revised) manuscript. We are glad to hear that we addressed your concerns. We believe your comments and suggestions improved the consistency and quality of our submission, and we greatly appreciate your contributions to this process.

---

### Official Review · Reviewer_qw6T · 2024-11-04

**Soundness:** 3
**Presentation:** 3
**Contribution:** 3
**Rating:** 8
**Confidence:** 2

**Summary:**

In this work, authors propose a new loss regularization term to reduce overconfidence in LLMs when training via the evidential deep learning (EDL) framework. This regularization is based on the information bottleneck loss, which maximizes mutual information between token logits and ground truth prediction distribution while minimizing M.I. between predictions and inputs. This is achieved by minimizing a KL divergence between a learned conditional logit distribution and a Gaussian prior. This work demonstrates that fine-tuning via this loss produces better calibration (ECE, AUC) with equivalent accuracy to baselines. Authors also illustrate this method is robust to noise in the training data and distribution shifts.

**Strengths:**

* This work is well-motivated theoretically, pointing out deficiencies with previous EDL losses (e.g. paragraph *Challenges when applying IB to an internal layer of an LLM*).
* The final regularization term $\mathcal{L}_{IB-Info}$ seems relatively simple, although I do have some questions about practical implementation (see questions)
* Strong experimental results; the OOD detection and noise injection experiments (tables 3,4) in particular seem support the argument for this additional regularization during calibration

**Weaknesses:**

* This work seems only applicable for a discrete set of mutually exclusive classes. I'm not sure how this extends to the case of open-ended generation where multiple responses may be semantically equivalent or have different logical relationships.
* While experiments show improvements in OOD detection when moving from OBQA--> ARC,CSQA datasets, I'd be interested to see the performance when the datasets are 'futher apart' semantically (e.g. moving from a reading comprehension to a math task).

**Questions:**

* While it appears the training methods/baselines tested are normalized for number of training steps, I am curious about comparison in terms of the number of training flops. Do the new mean and covariance prediction heads require separate training?
* Is there a benefit to having two parameters estimating the distributional shift (mean and variance) of the token predictions vs. just one?
* What happens to calibration if we finetune on multiple datasets simultaneously?
* How does this fine-tuning procedure effect LLM performance on open-ended generation tasks?

---

> ### Author Response · Authors · 2024-11-17
> **Response to Reviewer qw6T (1/2)**
>
> We thank the reviewer for the constructive comments and thoughtful suggestions. We address the concerns as follows:
>
> >W1: This work seems only applicable for a discrete set of mutually exclusive classes. I'm not sure how this extends to the case of open-ended generation where multiple responses may be semantically equivalent or have different logical relationships.
>
> **Theoretically, our method is applicable to both multi-choice tasks and open-ended generation.** This is because fine-tuning the model for open-ended generation involves predicting tokens within the vocabulary, corresponding to a case where the number of classes equals the vocabulary size. This scenario is already captured by our theoretical framework, so there is no theoretical barrier here. However, as we discussed in Section 6, we did not test our method on generative tasks due to the lack of well-established evaluations for model calibration in generative tasks. This remains an active research topic [1]. Testing IB-EDL on open-ended generation is indeed an exciting direction for future work. In this paper, addressing the overconfidence of LLMs in multi-choice tasks represents a significant step towards reliable LLM applications.
>
>
> >W2: While experiments show improvements in OOD detection when moving from OBQA--> ARC,CSQA datasets, I'd be interested to see the performance when the datasets are 'futher apart' semantically (e.g. moving from a reading comprehension to a math task).
>
> To address this concern, we evaluate the performance on the MMLU-Math [3] dataset (covering topics such as “college_mathematics”, “high_school_mathematics”, and “abstract_algebra”) as the OOD test set. We use OBQA as the in-distribution dataset, consistent with Table 3 of the paper. Additionally, we employ Llama2-7B for this experiment.
>
> As shown in Table E, we observe that:
> (i) IB-EDL consistently achieves the best OOD detection performance.
> (ii) In general, the AUROCs of our method and other baselines improve as the distribution shift increases.
>
> ### Table E: OOD Detection AUROC on OBQA -> MMLU-Math
> | Method  | Max. Probability (MP) $\uparrow$ | Uncertainty Mass (UM) $\uparrow$ |
> |--------|------------------------|------------------------|
> | MAP    | 91.36 ± 0.57          | -                      |
> | MCD    | 90.85 ± 0.33          | -                      |
> | Ens    | 90.68 ± 0.80          | -                      |
> | LA     | 91.09 ± 0.41          | -                      |
> | EDL    | 92.78 ± 0.26          | 92.86 ± 0.21           |
> | VID    | 91.64 ± 0.79          | 66.61 ± 4.98           |
> | I-EDL  | 91.48 ± 0.72          | 90.67 ± 0.88           |
> | R-EDL  | 88.44 ± 2.11          | 88.22 ± 1.70           |
> | IB-EDL (Ours) | **93.63 ± 0.66**      | **93.64 ± 0.56**       |
>
> >Q1: While it appears the training methods/baselines tested are normalized for number of training steps, I am curious about comparison in terms of the number of training flops. Do the new mean and covariance prediction heads require separate training?
>
> FLOPs are typically measured at inference time. As shown in Section 4.5, IB-EDL introduces only 1.98% additional FLOPs compared to the pre-trained LLM. To further address the reviewer’s question, we measure the number of samples processed per second as an indicator of computational complexity per step. Using Llama3-8B on a single NVIDIA H100 GPU, Table A shows that **the training and inference speeds of IB-EDL are comparable to those of MAP and other EDL methods.**
>
> The mean and covariance prediction heads are initialized using the pre-trained LLM’s linear head and equipped with LoRA adapters. Both are fine-tuned jointly with the transformer layers (i.e., backbone). Thank you for raising this point, and we will include these details in the revised manuscript.
>
> ### Table A: Additional Complexity Analysis
>
> | Method            | Test Samples/s | Training Samples/s | Training Time Memory (GB) |
> |-------------------|----------------|--------------------|---------------------------|
> | MAP               | 69.55 ± 2.86   | 26.57 ± 1.96       | 21.21 ± 0.35              |
> | MCD (10 forwards) | 9.79 ± 1.21    | -                  | -                         |
> | Ens (3 models)    | 25.77 ± 3.54   | -                  | -                         |
> | LA                | 5.95 ± 0.49    | -                  | -                         |
> | EDL               | 68.99 ± 1.59   | 26.44 ± 1.11       | 21.23 ± 0.15              |
> | VID               | 69.17 ± 0.99   | 26.69 ± 1.37       | 21.29 ± 0.37              |
> | I-EDL             | 68.94 ± 2.18   | 26.02 ± 0.71       | 21.33 ± 0.11              |
> | R-EDL             | 68.84 ± 1.09   | 26.47 ± 1.09       | 21.27 ± 0.21              |
> | IB-EDL  (Ours)          | 68.08 ± 1.75   | 26.41 ± 1.04       | 21.88 ± 0.66              |

---

> ### Author Response · Authors · 2024-11-17
> **Response to Reviewer qw6T (2/2)**
>
> >Q2: Is there a benefit to having two parameters estimating the distributional shift (mean and variance) of the token predictions vs. just one?
>
> For implementation, we employ a single linear head with double the output neurons, instead of two separate linear modules. This design allows us to predict the mean and variance in a single forward pass. As discussed in Q1, both parts of the common linear head are initialized with the pre-trained LLM’s linear head.
>
> Note that if we only estimate a mean parameter, we would have to assume an equal variance across different fine-tuning tasks, which would be a rather restrictive assumption.
>
>
> >Q3: What happens to calibration if we finetune on multiple datasets simultaneously?
>
> This is an intriguing question. Incorporating multiple datasets during fine-tuning reduces the likelihood of the model overfitting to spurious correlations in a single dataset, thereby potentially improving calibration. While this perspective has not been extensively explored in prior work, we believe that enhancing calibration by increasing the diversity of training data represents an interesting and promising direction for future research.
>
> >Q4: How does this fine-tuning procedure effect LLM performance on open-ended generation tasks?
>
> As discussed in response to W1, this question remains open and warrants further exploration. The answer will become clearer as the community develops well-established metrics for evaluating calibration and uncertainty in open-ended generation tasks. Testing IB-EDL under these scenarios is an exciting future direction.
>
> ---
>
> Thank you again for your insightful comments. We will incorporate your suggestions in a revised version of the manuscript. We warmly invite you to engage in further interactive discussions or share additional suggestions.
>
>
> [1] Kuhn, Lorenz, Yarin Gal, and Sebastian Farquhar. "Semantic Uncertainty: Linguistic Invariances for Uncertainty Estimation in Natural Language Generation." The Eleventh International Conference on Learning Representations.

---

> ### Author Response · Authors · 2024-11-24
> **The end of the rebuttal phase is approaching**
>
> Dear Reviewer qw6T,
>
> Thank you for your thoughtful review and constructive feedback on our manuscript. We greatly appreciate your recognition of the theoretical motivation and the strong experimental results.
>
> As the discussion period will conclude in approximately 72 hours, we would like to kindly check whether our responses thus far have successfully addressed your concerns and clarified your questions.
>
> In summary, we addressed the key points raised by you as follows:
>
> * **OOD detection on semantically distant datasets:** We added experiments evaluating OOD detection performance on the MMLU-Math dataset (see Table E), demonstrating that IB-EDL achieves superior OOD performance, even with increased semantic distance.
>
> * **Training FLOPs and head design:** We detailed in Table A that IB-EDL introduces minimal additional computational overhead and that the mean and variance are predicted via a single linear head in one forward pass, ensuring efficiency.
>
> If there are any additional questions or remaining concerns, we would be happy to address them.
>
> Should you find that our responses sufficiently address your feedback, we would sincerely appreciate your consideration of raising the score, if deemed appropriate.
>
> Thank you again for your valuable input and for helping us improve our work.
>
> Best regards,
>
>
> Authors of IB-EDL

---

> ### Comment · Reviewer_qw6T · 2024-11-27
> **Acknowledgment of Author Response**
>
> Thank you for the detailed response and additional experiments. Overall I think the reviewer's initial response addresses most of my concerns and I am willing to raise my score from marginal accept to accept.
>
> * The detailed distribution shift experiments (including those in response to reviewer Tujz) help to demonstrate the benefits of this method over previous baselines and satisfy my concerns about the dependence on 'semantic distance' between sets.
> * Thank you for the clarification on training of the mean/covariance heads and additional results for wall-clock time of training.
> * While I agree that this method can possibly be extended to a discrete set of token sequences, the number of token sequences in open-ended generation is infinite and by definition the dirichlet prior is over discrete distributions. Additionally, it seems that y is assumed to be a one-hot vector, while in reality multiple semantically equivalent token sequences may be correct for a single question. I would like to see a bit more nuanced discussion on this extension but agree that experimentally it can be left to future work.

---

> > ### Author Response · Authors · 2024-11-27
> > **Follow-up discussion**
> >
> > Dear Reviewer qw6T,
> >
> > Thank you for your thoughtful feedback and for acknowledging our responses. We are pleased that we addressed your concerns.
> >
> > Regarding the follow-up discussion on extending our approach to open-ended generation, we find this to be an intriguing and important question.
> >
> > Here, we share our thoughts on the question, *"How will this fine-tuning procedure affect performance on open-ended generation tasks, where semantic equivalence exists?"*.
> >
> > To provide a comprehensive response, we have structured our thoughts as follows: We first check whether EDL can still be applied to training in the presence of semantic equivalence. Then, we share our conjecture about how EDL could affect semantic equivalence in generated outputs. Lastly, we briefly discuss how other papers treat semantic equivalence when measuring uncertainty or calibration error in the generation tasks.
> >
> > These points are detailed as follows:
> >
> > * **Dirichlet priors remain applicable in generation scenarios.** While LLMs can produce many different sequences, the individual tokens in these sequences are drawn from a fixed and finite vocabulary. Moreover, LLMs generate sequences in an auto-regressive manner, where the model uses previously generated tokens as input to predict the next token at each step. While the generation of new text has seemingly infinitely many possibilities, the next token given the previously generated sequence follows a finite Categorical distribution. For this categorical distribution, a Dirichlet serves as a natural prior and can still be applied. **Importantly, the Dirichlet prior is applied at the token level, not at the sequence level.** This ensures its compatibility with the auto-regressive framework of LLMs.
> >
> > * **Semantic equivalence does not impede the applicability of EDL to LLM training.**  While open-ended generation tasks often involve multiple semantically equivalent token sequences as valid answers, training still operates on individual question-answer pairs. For instance, given the question "6 + 8 = ?" and answers "14" or "fourteen", training pairs would be structured as separate samples: ("6 + 8 = ?", "14") and ("6 + 8 = ?", "fourteen"), instead of a triplet ("6 + 8 = ?", "14", "fourteen"). Each pair retains a single ground truth encoded as a one-hot sequence. Consequently, the EDL pipeline and the Subjective Logic framework remain applicable.
> >
> > * **EDL methods may encourage diverse yet semantically equivalent outputs.** Our conjecture is based on our observation that IB-EDL, R-EDL and I-EDL promote smoother token probability distributions compared to standard MLE training. This smoothing effect could increase the likelihood of assigning similar probabilities to tokens that produce semantically equivalent outputs (e.g., "14" and "fourteen"). With techniques like Beam Search or Nucleus Sampling being applied during inference, models trained with EDL could generate more diverse yet semantically equivalent answers. However, this is a conjecture, and a rigorous analysis would require sensible metrics to quantify calibration error or uncertainty in generation tasks. We will discuss some existing metrics in the subsequent paragraph.
> >
> > * **Addressing semantic equivalence through metrics in semantic space.** Several recent works agree that semantic equivalence should be considered when evaluating uncertainty [1, 2] or calibration [3] in generative tasks.  These works propose embedding generated outputs into a continuous (semantic) space using a secondary language model, followed by clustering semantically equivalent outputs into clusters. Calibration error or uncertainty is then measured at the cluster level. More concretely, when measuring calibration error, the authors of [3] define the bins of ECE based on these semantic clusters rather than using probability thresholds. In summary, these evaluation metrics are an ongoing topic. We believe that, as these evaluation methods advance, they will enable more precise assessments of the impact of EDL on generative tasks.
> >
> > We would greatly appreciate it if you could share your thoughts on this additional topic. We will incorporate this discussion into a future revision of the manuscript and thoughtfully integrate your feedback.
> >
> > Lastly, we would sincerely appreciate your consideration of raising the score to further support our work😊.
> >
> > Thanks again for your guidance throughout the reviewing process.
> >
> > ---
> >
> > [1] Kuhn, Lorenz, Yarin Gal, and Sebastian Farquhar. "Semantic uncertainty: Linguistic invariances for uncertainty estimation in natural language generation." arXiv preprint arXiv:2302.09664(2023).
> >
> > [2] Nikitin, Alexander, et al. "Kernel Language Entropy: Fine-grained Uncertainty Quantification for LLMs from Semantic Similarities." arXiv preprint arXiv:2405.20003 (2024).
> >
> > [3] Ulmer, Dennis, et al. "Calibrating Large Language Models Using Their Generations Only." arXiv preprint arXiv:2403.05973 (2024).

---

### Official Review · Reviewer_tujz · 2024-11-04

**Soundness:** 3
**Presentation:** 3
**Contribution:** 2
**Rating:** 6
**Confidence:** 3

**Summary:**

This paper proposes IB-EDL, combining Evidential Deep Learning (EDL) with information bottleneck (IB) for uncertainty estimation in large language models (LLMs). This approach aims to enhance model calibration by reducing overconfidence, which is a common issue in fine-tuned LLMs. The experimental results show that IB-EDL achieves improved calibration with minimal computational overhead.

**Strengths:**

The paper is well-presented and easy to follow.

The paper proposes a unified perspective on EDL methods by framing several existing EDL methods as special cases within the IB-EDL framework.

The experimental results are strong and covers multiple LLMs, showing that IB-EDL addresses the overconfidence problem well.

**Weaknesses:**

The pipeline of applying IB to EDL ($x → f(x; θ) → e˜ → e →
α → π → y$) seems computational heavy, it would be informative to include an inference time comparison of IB-EDL against other methods.

Although the training overhead of IB-EDL is relatively small compared with the pretraining step, the paper lacks comparison of training time compared with LoRA, which is the fine-tuning backbone of IB-EDL.

For out-of-distribution (OOD) scenarios, the paper only compares the OOD detection capabilities with AUROC. The expected calibration error (ECE) is missing, which is also an indicator of OOD performance as the calibrated model should maintain relatively low ECE score on OOD datasets.

The novelty is not clear for combining IB with EDL, which are two well established methods.

**Questions:**

Please see weakness.

---

> ### Author Response · Authors · 2024-11-17
> **Response to Reviewer tujz (1/3)**
>
> We thank the reviewer for the constructive comments and feedback. We address the concerns as follows:
>
> >W1: The pipeline of IB-EDL ( $x \rightarrow f(x; \theta) \rightarrow \tilde{e} \rightarrow e \rightarrow \alpha \rightarrow \pi \rightarrow y$) seems computationally heavy, it would be informative to include an inference time comparison of IB-EDL against other methods.
>
> Compared to other baselines such as MAP, the additional complexity of IB-EDL lies in the augmented final linear layer. However, the main computational cost of the model is induced by the transformer layers rather than the linear head. Furthermore, after the forward pass through the transformer layers and the final linear layer, the subsequent steps in the pipeline ($\tilde{e} \rightarrow e \rightarrow \alpha \rightarrow \pi \rightarrow y$) are basic algebraic operations and do not involve any model parameters.
>
> In Section 4.5, we present a complexity analysis showing that IB-EDL adds less than 2% extra FLOPs to the pre-trained model. In addition, we now include more detailed training/test time and memory analyses. Specifically, we conducted experiments with Llama3-8B on the OBQA dataset using a single NVIDIA H100 GPU. As shown in Table A, **IB-EDL’s training and inference speeds are comparable to those of the MAP and EDL baselines and are significantly faster than MCD or Ens**. Additionally, the extra memory cost of IB-EDL is also marginal.
>
> Table A: Additional Complexity Analysis
> | Method            | Test Samples/s | Training Samples/s | Training Time Memory (GB) |
> |-------------------|----------------|--------------------|---------------------------|
> | MAP               | 69.55 ± 2.86   | 26.57 ± 1.96       | 21.21 ± 0.35              |
> | MCD (10 forwards) | 9.79 ± 1.21    | -                  | -                         |
> | Ens (3 models)    | 25.77 ± 3.54   | -                  | -                         |
> | LA                | 5.95 ± 0.49    | -                  | -                         |
> | EDL               | 68.99 ± 1.59   | 26.44 ± 1.11       | 21.23 ± 0.15              |
> | VID               | 69.17 ± 0.99   | 26.69 ± 1.37       | 21.29 ± 0.37              |
> | I-EDL             | 68.94 ± 2.18   | 26.02 ± 0.71       | 21.33 ± 0.11              |
> | R-EDL             | 68.84 ± 1.09   | 26.47 ± 1.09       | 21.27 ± 0.21              |
> | IB-EDL  (Ours)          | 68.08 ± 1.75   | 26.41 ± 1.04       | 21.88 ± 0.66              |
>
> >W2: Although the training overhead of IB-EDL is relatively small compared with the pretraining step, the paper lacks comparison of training time compared with LoRA, which is the fine-tuning backbone of IB-EDL.
>
> To address this concern, it is sufficient to compare IB-EDL and MAP in Table A. This is because the MAP setting corresponds exactly to LoRA fine-tuning of the pre-trained LLM. As Table A demonstrates, IB-EDL does not show a significant slowdown in training time compared to MAP.

---

> ### Author Response · Authors · 2024-11-17
> **Response to Reviewer tujz (2/3)**
>
> >W3: For out-of-distribution (OOD) scenarios, the paper only compares the OOD detection capabilities with AUROC. The expected calibration error (ECE) is missing, which is also an indicator of OOD performance as the calibrated model should maintain relatively low ECE score on OOD datasets.
>
> We now present additional calibration experiments in Table B, C, and D. These tables show that IB-EDL achieves the best calibration performance on two out of three OOD datasets. This result demonstrates that **IB-EDL’s calibration capabilities generalize well under distribution shifts.**
>
> ### Table B: Calibration in the setting OBQA -> ARC-C
> | Method | Acc              | ECE             | NLL             |
> |--------|------------------|-----------------|-----------------|
> | MAP    | 79.18 ± 0.45     | 18.04 ± 0.38    | 1.30 ± 0.02     |
> | MCD    | 79.16 ± 0.43     | 18.06 ± 0.36    | 1.33 ± 0.06     |
> | Ens    | 79.27 ± 0.45     | 16.74 ± 0.13    | 1.19 ± 0.03     |
> | LA     | **79.38 ± 0.40** | 6.61 ± 0.41     | 0.73 ± 0.02     |
> | EDL    | 78.27 ± 0.79     | 7.65 ± 0.47     | 0.74 ± 0.01     |
> | VID    | 78.27 ± 0.57     | 8.74 ± 0.40     | 0.78 ± 0.01     |
> | I-EDL  | 78.55 ± 0.30     | 7.68 ± 1.19     | 0.73 ± 0.02     |
> | R-EDL  | 78.32 ± 1.24     | 5.03 ± 0.62     | 0.73 ± 0.04     |
> | IB-EDL (Ours) | 78.31 ± 1.14     | **4.67 ± 1.09** | **0.72 ± 0.02** |
>
> ### Table C: Calibration in the setting OBQA -> ARC-E
> | Method | Acc              | ECE             | NLL             |
> |--------|------------------|-----------------|-----------------|
> | MAP    | 88.06 ± 0.20     | 9.63 ± 0.47     | 0.71 ± 0.03     |
> | MCD    | 88.05 ± 0.20     | 9.63 ± 0.46     | 0.70 ± 0.04     |
> | Ens    | 88.15 ± 0.02     | 9.19 ± 0.65     | 0.68 ± 0.05     |
> | LA     | **88.36 ± 0.24** | **3.02 ± 0.15** | **0.42 ± 0.02** |
> | EDL    | 86.38 ± 0.87     | 10.11 ± 0.85    | 0.52 ± 0.01     |
> | VID    | 87.41 ± 0.82     | 3.19 ± 0.19     | 0.46 ± 0.02     |
> | I-EDL  | 87.55 ± 0.20     | 11.83 ± 0.65    | 0.53 ± 0.01     |
> | R-EDL  | 87.31 ± 0.93     | 5.32 ± 0.46     | 0.46 ± 0.01     |
> | IB-EDL (Ours) | 87.94 ± 0.22     | 5.03 ± 0.15     | 0.44 ± 0.02     |
>
> ### Table D: Calibration in the setting OBQA -> CSQA
> | Method | Acc              | ECE             | NLL             |
> |--------|------------------|-----------------|-----------------|
> | MAP    | 69.37 ± 0.67     | 27.85 ± 0.53    | 2.06 ± 0.04     |
> | MCD    | 69.38 ± 0.68     | 27.86 ± 0.46    | 2.13 ± 0.13     |
> | Ens    | 69.14 ± 0.47     | 27.07 ± 1.32    | 2.01 ± 0.10     |
> | LA     | 69.34 ± 0.58     | 11.99 ± 0.72    | 1.15 ± 0.01     |
> | EDL    | 69.34 ± 0.88     | 8.32 ± 1.33     | 0.98 ± 0.03     |
> | VID    | 69.99 ± 1.07     | 16.66 ± 0.73    | 1.06 ± 0.02     |
> | I-EDL  | 70.49 ± 0.56     | 5.96 ± 0.71     | 0.94 ± 0.01     |
> | R-EDL  | 70.62 ± 1.28     | 12.84 ± 1.73    | 1.00 ± 0.05     |
> | IB-EDL (Ours) | **71.29 ± 0.96** | **4.51 ± 0.15** | **0.93 ± 0.02** |

---

> ### Author Response · Authors · 2024-11-17
> **Response to Reviewer tujz (3/3)**
>
> >W4: The novelty is not clear for combining IB with EDL, which are two well established methods.
>
> Our work is not a simple combination of these methods; instead, **it tackles key unresolved issues in EDL (see also [3, 4]) and addresses the challenge when regularizing EDL with IB**. Importantly, implementing IB on EDL in a naïve manner, such as inserting IB at an arbitrary layer, can compromise theoretical soundness. Our contributions go beyond mere implementation and address these challenges through the following advancements:
>
> 1. **New theoretical insight:** We uncover a critical issue overlooked by previous variational IB literature [1, 2]: applying IB at an arbitrary internal LLM layer can result in an optimization objective that is no longer a valid lower bound of $I(Z, Y)$. In Section 3.1, we provide a detailed theoretical analysis through Eq. (11).
>
> 2. **Novel solution:** We propose a unique and novel solution to the identified challenge. We show that, in EDL, this challenge can be addressed by selecting the pre-evidence $\tilde{e}$ as the IB variable. This approach ensures that the optimization objective remains a valid variational bound. Furthermore, as Eq. (14) illustrates, our IB method introduces L2 regularization on the evidence, directly preventing the model from producing extremely large evidence values and thereby reducing overconfidence. This solution directly addresses the drawbacks of EDL identified in prior works [3, 4].
>
> 3. **Unified perspective:** Through Proposition 1, we unify previous approaches, providing a cohesive theoretical framework. This unified viewpoint is also novel and provides valuable insights into the family of EDL methods.
>
>
> In addition to these theoretical contributions, we also offer practical advancements. For instance, previous EDL works only validated their methods on ResNet-sized models. In contrast, we demonstrate that **our EDL method scales to billion-parameter-sized LLMs**. This advancement opens new avenues for extending EDL methods to broader research and application domains.
>
> ---
>
> Thank you again for the insightful comments and we will incorporate your suggestions in a revised version of the manuscript. We warmly invite you to engage in further interactive discussions to explore any additional perspectives you might have.
>
>
> [1] Alemi, Alexander A., et al. "Deep variational information bottleneck." arXiv preprint arXiv:1612.00410 (2016).
>
> [2] Wieczorek, Aleksander, and Volker Roth. "On the difference between the information bottleneck and the deep information bottleneck." Entropy 22.2 (2020): 131.
>
> [3] Deng, Danruo, et al. "Uncertainty estimation by fisher information-based evidential deep learning." International Conference on Machine Learning. PMLR, 2023.
>
> [4] Chen, Mengyuan, Junyu Gao, and Changsheng Xu. "R-EDL: Relaxing Nonessential Settings of Evidential Deep Learning." The Twelfth International Conference on Learning Representations.

---

> ### Author Response · Authors · 2024-11-24
> **The end of the rebuttal phase is approaching**
>
> Dear Reviewer tujz,
>
> Thank you for your thorough review and constructive feedback on our manuscript. We greatly appreciate your recognition of the strengths of our work, including the strong experimental results, the clear presentation, and the unified theoretical perspective that IB-EDL provides for EDL research.
>
> As the discussion period is set to conclude in approximately 72 hours, we would like to kindly check whether our responses thus far have successfully addressed your concerns and clarified your questions.
>
> In summary, we addressed the points raised by you as follows:
>
> * **Inference and training time comparisons:** We provided detailed complexity analyses (Section 4.5 in paper and Table A here), showing that IB-EDL introduces minimal computational overhead during training and inference, with performance comparable to MAP and EDL baselines.
>
> * **Comparison with LoRA training time:** We clarified that the MAP setting corresponds directly to LoRA fine-tuning, and our experiments confirm that IB-EDL incurs negligible additional overhead compared to LoRA.
>
> * **Calibration on OOD datasets:** We conducted additional experiments to evaluate ECE on OOD datasets. As shown in Tables B, C, and D, IB-EDL achieves strong calibration performance under distribution shifts.
>
> * **Novelty of IB-EDL:** We highlighted that IB-EDL is not a simple combination of existing methods but addresses critical theoretical challenges in applying IB to EDL, as detailed in Section 3.1 and Section 3.2. We also provided a unified framework for EDL methods and demonstrated scalability to large LLMs, which represents a significant advancement over prior works.
>
> If there are any additional questions or concerns, we would be happy to address them.
>
> Should you find that our responses sufficiently address your feedback, we would sincerely appreciate your consideration of raising the score, if deemed appropriate.
>
> Best regards,
>
> Authors of IB-EDL

---

> ### Comment · Reviewer_tujz · 2024-11-30
>
> Thanks for addressing my concerns, I have raised the score accordingly.

---

> > ### Author Response · Authors · 2024-11-30
> > **Thank you**
> >
> > Dear Reviewer tujz,
> >
> > Thank you for your thoughtful review and constructive feedback, which have been invaluable in helping us improve the quality of our work. We are glad that we have addressed your concerns, and we truly appreciate your positive assessment.
> >
> > Best regards,
> >
> > Authors

---

### Official Review · Reviewer_TTkc · 2024-11-04

**Soundness:** 4
**Presentation:** 3
**Contribution:** 3
**Rating:** 8
**Confidence:** 4

**Summary:**

The paper present a method to effectively apply Evidential Deep Learning (EDL), a parametric uncertainty estimation approach, to LLMs. The approach introduces two features to standard EDL that make it practical and effective for application to LLMs; i) the introduction of an information bottleneck cost function, which mitigates over-fitting and ii) the use of the LLM outputs as the intermediate "layer" Z where to impose the bottleneck, which simplifies direct computation of certain terms in the mutual information lower bounds, part of the objective function.

**Strengths:**

- Relevance: The paper addresses an important problem; the uncertainty calibration of LLM outputs (at the token level), which is an important component for providing safe deployment and error estimation for LLMs.

- Theoretical Soundness: The paper takes a very principled approach, identifying precise issues in the application of EDL to LLMs and addressing them with mathematically derived solutions based on reasonable starting assumptions.

- Experimental Soundness: The experiments are more than sufficiently extensive: they target the right settings and metrics to validate the claims of the paper and comprehend great breadth of models, tasks and mode of evaluation (acc+ECE, OOD detection). The claims of improved uncertainty calibration are strongly backed by the experiments.

**Weaknesses:**

The only main weakness of the paper is the clarity of scope:

From the experiments, it is clear that this is a method for uncertainty calibration for a specific task at the fine-tuning stage of an LLM. e.g., an LLM is fine-tuned for summarisation and the proposed method offers a way to obtain superior uncertainty calibration *for the fine-tuned task*. This is indeed very useful, but needs to be stated more clearly in the abstract and intro, as from all sections up until the experiments it is unclear if this is a method for pre-training and/or fine-tuning. In principle, I can see the method may be applied to pre-training too, but since this is not tested in the experiments it should not be included in the claims.

**Questions:**

See and clarify the weakness point above.

---

> ### Author Response · Authors · 2024-11-17
> **Response to Reviewer TTkc**
>
> We sincerely thank the reviewer for recognizing our contributions and for the constructive feedback. We address the concerns as follows:
>
> > W1.1: Scope of the work.
>
> We thank the reviewer for pointing this out. Our work focuses on calibrating *fine-tuned* LLMs and enhancing their uncertainty awareness. We agree that emphasizing this focus will improve the clarity of our work. We will explicitly emphasize the scope of our work by revising the first bullet point in the list of contributions in the paper's introduction to “We introduce IB-EDL, an information-theoretic approach to improve calibration in fine-tuned LLMs.
>
>
> > W1.2: Applicability of IB-EDL to pre-training.
>
> IB-EDL can theoretically be applied to pre-training. However, we conducted experiments on fine-tuning LLMs due to two primary reasons: (i) pre-training LLMs demands immense computational resources and has a substantial carbon footprint, and (ii) compared to pre-trained LLMs, fine-tuned LLMs present a more pressing need for improved calibration [1, 2]. That said, we agree that exploring the application of IB-EDL to pre-training represents an exciting and valuable avenue for future research. We will add a corresponding sentence to our conclusion. We thank the reviewer for highlighting this potential extension of our work.
>
> ---
>
> We thank the reviewer again for the constructive feedback. Please let us know if our response addresses the concerns and if there is anything else we need to clarify.
>
> [1] Yang, Adam X., et al. "Bayesian Low-rank Adaptation for Large Language Models." The Twelfth International Conference on Learning Representations.
>
> [2] Achiam, Josh, et al. "Gpt-4 technical report." arXiv preprint arXiv:2303.08774 (2023)

---

> ### Author Response · Authors · 2024-11-24
> **The end of the rebuttal phase is approaching**
>
> Dear Reviewer TTkc,
>
> Thank you once again for your thoughtful and constructive feedback on our manuscript. We are truly encouraged by your recognition of the soundness of our theoretical approach, the relevance of our problem, and the robustness of our experimental results.
>
> As the discussion period will conclude in approximately 72 hours, we wanted to check if our responses thus far have adequately addressed your concerns and clarified your questions.
>
> In summary, we have clarified the scope of our method and that it is specifically designed to improve uncertainty calibration in fine-tuned LLMs. We have revised our introduction to explicitly emphasize this focus, ensuring the scope is clear from the outset.
>
> If there are any remaining concerns or questions, we would be happy to address them.
>
> Best regards,
>
> Authors of IB-EDL

---

### Author Response · Authors · 2024-11-22
**Key Updates of Manuscript**

**Dear Reviewers**,

As the deadline for the discussion phase approaches, we have submitted an improved manuscript incorporating your constructive comments and feedback. Below, we summarize the key updates made:

### Key Updates

**Additional Experiments**: We appreciate your recognition of the extensive experiments included in our original submission. Building on this foundation and incorporating your valuable feedback, we have added the following additional experiments: (1) OOD detection under large distribution shifts, (2) Calibration performance on OOD datasets, (3) Analysis of training and inference speed, as well as memory consumption, (4) Sensitivity analysis of the number of ECE bins.

These updates are detailed in Appendix D.2–D.6, with corresponding references and discussions added in the main text.

**Highlighting Novelty**: To better articulate our contributions, we have revised the introduction to emphasize the novelty of our approach. Specifically:

Our method goes beyond a straightforward implementation of IB on EDL. Importantly, it introduces several key advancements:

* **New theoretical insights**: We uncover a theoretical issue overlooked in previous IB literature.

* **Innovative solution**: We address this issue in the context of EDL by altering the IB's latent variable parametrization.

* **Effective regularization**: Using regularization on the evidence, we mitigate the problem of overconfident evidence values reported in previous EDL studies.

* **Unified framework**: Our IB-EDL framework unifies several previous EDL methods, offering a comprehensive perspective and advancements to the field.

* **Scalability to LLMs**: We scale EDL methods to LLMs, extending their applicability beyond the medium-sized neural networks traditionally used in EDL literature. This significantly enhances the potential for developing reliable, modern large-scale applications.

**Additional Related Work**: In response to Reviewer 4Vtx, we have expanded the related work section to include MixLoRA.

**Clarification of Research Scope**: As suggested by Reviewer TTkc, we have revised the introduction to clarify the research scope and focus of our work.

---

If you have any further concerns or suggestions, please do not hesitate to let us know. We deeply appreciate your valuable insights and guidance throughout the review process. Thank you again for your time and effort in reviewing our work.

---

### Meta-Review · Area_Chair_23Fz · 2024-12-19

**Metareview:**

The paper present a method to apply evidential deep learning, a parametric uncertainty estimation approach, to LLMs. The proposed approach introduces an information bottleneck cost function over the LLM outputs as well as the model parameter. The reviewers raised points about the applicability to more general settings, computataional complexity, and clarity of the presentation. During the rebuttal, the authors have addressed these questions by revising the paper and by providing additional experiments. All the reviewers are positive about the paper, after the rebuttal period.

**Additional Comments On Reviewer Discussion:**

The reviewers raised points about the applicability to more general settings, computataional complexity, and clarity of the presentation. During the rebuttal, the authors have addressed these questions by revising the paper and by providing additional experiments.

---

### Decision · Program_Chairs · 2025-01-22

Accept (Poster)